



# Impact of bottom trawling on sediment biogeochemistry: a modelling approach

Emil De Borger[1,2], Justin Tiano[2,1], Ulrike Braeckman[1], Adriaan D. Rijnsdorp[3], Karline Soetaert[2,1].

[1]Ghent University, Department of Biology, Marine Biology Research Group, Krijgslaan 281/S8, 9000 Ghent, Belgium
[2]Royal Netherlands Institute of Sea Research (NIOZ), Department of Estuarine and Delta Systems, and Utrecht University, Korringaweg 7, P.O. Box 140, 4401 NT Yerseke, The Netherlands
[3]Wageningen Marine Research, Wageningen University & Research, IJmuiden, The Netherlands

*Correspondence to* : Emil De Borger (emil.de.borger@nioz.nl)

## Abstract

Bottom trawling in shelf seas can occur more than 10 times per year for a given location. This affects the benthic metabolism, through a mortality of the macrofauna, resuspension of organic matter from the sediment, and alterations of the physical sediment structure. However, the trawling impacts on organic carbon mineralization and associated processes are not well known. Using a modelling approach, the effects of increasing trawling frequencies on early diagenesis were studied in five different sedimentary environments, simulating the effects of a deep penetrating gear (e.g. a tickler chain beam trawl) and a

shallower, more variable penetrating gear (e.g. an electric pulse trawl). Trawling events strongly increased oxygen and nitrate concentrations in surface sediment layers, and led to significantly lower amounts of ammonium (43 – 99 % reduction) and organic carbon in the top 10 cm of the sediment (62 – 96 % reduction). As a result, total mineralization rates in the sediment were decreased by up to 28 %. The effect on different mineralization processes differed both between sediment types, and between trawling frequencies. The shallow penetrating gear had a slightly smaller effect on benthic denitrification than the

deep penetrating gear, but there were no statistically different results between gear types for all other parameters. Denitrification was reduced by 69 % in a fine sandy sediment, whereas nitrogen removal nearly doubled in a highly eutrophic mud. This suggests that even relatively low penetration depths from bottom fishing gears generates significant biogeochemical alterations. Physical organic carbon removal through trawl-induced resuspension of sediments, exacerbated by a removal of bioturbating macrofauna, was identified as the main cause of the changes in the mineralization process.

## 1   Introduction

Bottom trawl fisheries provide for 23% of global fish landings (Cashion et al., 2018), with the vast majority of this type of fishing taking place in productive coastal shelf seas (Amoroso et al., 2018). In bottom trawl fisheries, nets are dragged along the bottom with the help of weighted devices such as otter boards, shoes, or beams, while chains, groundropes, and/or electrical stimuli are used to coerce fish into the net. Bottom trawl gears penetrate the seafloor, up to 35 cm deep for otter trawl boards,





30 and 10 cm deep for tickler chain rigged beam trawls, depending on the gear specifics and the sediment type (Eigaard et al., 2016). Hence, during a trawling event, sediment is mixed down to a certain depth, and hydraulic drag introduced by the moving gear can cause the erosion of an additional sediment layer (Depestele et al., 2016, 2019; O'Neill and Summerbell, 2011; O'Neill and Ivanović, 2016). Sediment disturbances by bottom trawling occur on very large scales: 63% of all North Sea sediments are trawled between 1 and more than 10 times per year (Eigaard et al., 2017).

35 Scientific literature is rich in studies showing the physical and ecological alterations to the benthic environment caused by bottom trawling. Acute impacts of bottom trawling include the homogenization of surface sediment (Depestele et al., 2019; Ferguson et al., 2020) and the removal of significant proportions of benthic fauna (Bergman and Hup, 1992; Bergman and Van Santbrink, 2000, Tiano et al., 2020). Consistent fishing pressure favors organisms with shorter life spans and/or increasing resistance to trawling, while communities become depleted of species with key functional roles (Kaiser et al., 2006; Hiddink 40 et al., 2017; Sciberras et al., 2018). Both fining (Trimmer et al., 2005) and coarsening (Palanques et al., 2014; Mengual et al., 2016) of the sediment has been attributed to trawling, as well as chronic organic matter depletion (Pusceddu et al., 2014; Paradis et al., 2019). The effects of bottom trawling on biogeochemical dynamics, however, remain relatively understudied. Trawling has been linked with enhanced carbon mineralization rates due to organic matter priming (van de Velde et al., 2018) and/or trawl-induced increases in organic material (Polymenakou et al., 2005; Pusceddu et al., 2005; Palanques et al., 2014; 45 Sciberras et al., 2016). These results seemingly contrast with findings of organic matter depletion (Mayer et al., 1991; Brylinsky et al., 1994; Watling et al., 2001) and reduced mineralization rates after acute trawling (Tiano et al., 2019), highlighting the lack of knowledge on this topic and the need for further investigation.

 Geochemical alterations impact the capacity of the sediment to recycle organic matter back to bioavailable nutrients (i.e. the sediment biogeochemistry). These are important processes in shallow coastal seas where primary production is strongly 50 dependent on nutrients regenerated in the sediments (Soetaert and Middelburg, 2009; Provoost et al., 2013). Observed biogeochemical changes caused by sediment resuspension can lead to the instantaneous release of nutrients from the sediment into the water column, temporarily enhanced oxygen consumption (Tiano et al., 2019) and increased nutrient concentrations in the bottom water (Riemann and Hoffmann, 1991; Almroth et al., 2009; Couceiro et al., 2013). Furthermore, trawling has been linked to an increase of the sediment oxygenated layer (Allen and Clarke, 2007; Tiano et al., 2019) and a reduction of the 55 denitrification capacity of cohesive sediments (Ferguson et al., 2020). It has been argued, based on *in situ* measurements, that the sediment biogeochemistry in consistently disturbed sediments remains in a transient state, i.e. the sediments are permanently? recovering from a disturbance event (Van De Velde et al., 2018). These effects can potentially be mitigated with alternative fishing gears or modified gear configurations, however, the effectiveness of which needs to be assessed.

 To reduce fishing impacts, alternative bottom trawl gears such as pulse fishing gears are being investigated (Van Marlen 60 et al., 2014; McConnaughey et al., 2020). With pulse gears, the heavy tickler chains are replaced by electrodes, which emit electrical pulses that induce a cramping response in flatfish (Soetaert et al., 2015). This causes fish to become temporarily immobilized, allowing their capture in a net, which drags behind the electrodes. Pulse gears exhibit lower penetration depths (~50%) than conventional beam trawls (Depestele et al., 2019), and also erode less material into suspension through





hydrodynamic drag due to a reduced towing speed (Rijnsdorp et al., 2020a). The lower penetration depth of the pulse gear
compared to standard tickler chain methods, has been shown to decrease the effects of bottom trawling on the sediment redox
layer (Depestele et al., 2019) and on chlorophyll *a* reduction (Tiano et al., 2019) but it is yet unclear how specific mineralization
processes might be affected on longer temporal scales.

The aim of this study was to explore the possible impacts of bottom trawling on the sediment carbon and nitrogen cycling
for two gears with different penetration depth distributions, and with increasing trawling frequency. We use a dynamic
diagenetic model, to which trawling disturbances were added. We parametrized the model for five locations in the North Sea,
with sediments ranging from coarse sands to fine mud. Our hypotheses were (1) that the effects of bottom trawling would
differ depending on the sedimentary environment, and (2) that fishing gear with reduced sediment penetration would incur less
changes in biogeochemical cycling.

## 2 Materials and methods

### 2.1 Model setup

#### 2.1.1 Model description.

To model the effects of bottom trawling on sediment biogeochemistry, disturbance events were added to a dynamic
implementation of the early diagenesis model OMEXDIA (Soetaert et al., 1996a, 1996b). This model describes the
concentrations of organic matter, oxygen, nitrate, ammonium, dissolved inorganic carbon (DIC) and oxygen demand units
(ODU's, reduced reaction products of anoxic mineralization). These are calculated on a 1D grid, with 100 layers increasing in
thickness, starting from 0.01 cm at the sediment water interface (SWI) and extending up to a sediment depth of 100 cm. The
incoming flux of organic matter (detritus) consists of a labile, fast decaying fraction (FDET) and a refractory, slow decaying
fraction (SDET), and is mineralized in either oxic mineralization, denitrification, or anoxic mineralization (Table 1). With oxic
mineralization and denitrification, the consumption of oxygen and nitrate as terminal acceptors is explicitly modelled (Table
1; Eqs. 1.1, 1.2). Anoxic mineralization processes with alternative oxidants such as manganese oxides, iron oxides, sulphate,
and organic matter are collected into one process that produces oxygen demand units (ODU's) as reaction products (Table 1;
Eq. 1.3). ODU reoxidation, and nitrification, the biological oxidation of ammonia to nitrate, are two additional processes that
consume oxygen (Table 1; Eqs. 1.4, 1.5). Mineralization rates are dependent on carbon availability (first order kinetics), and
oxidant availability (Michaelis-Menten type kinetics), and inhibited by concentrations of inhibiting solutes (e.g., oxygen
inhibits denitrification and anoxic mineralization). FDET, SDET, $O_2$, $NO_3^-$, $NH_4^+$, ODU's, and DIC are the 7 state variables
of which the concentrations are modelled in every layer.

Exchange of state variables between the different layers is caused by advection (sediment accretion, $v$) or molecular
diffusion (for solutes), and bioturbation (for solids). The solute flux $J_D$ due to molecular diffusion and advection is described
by Fick's first law (Fick, 1855),



$$J_D = \varphi D_i \frac{\partial C}{\partial z} + \varphi v C \qquad (Eq.\ 1)$$

where the effective diffusion coefficient is estimated as $D_i = D_0/\theta^2$, with $D_0$ the molecular diffusivity of the solute, $\theta^2 = 1 - 2\ln(\varphi)$ the factor correcting for sediment tortuosity (Boudreau, 1996), $\varphi$ the sediment porosity, which was kept constant with depth $z$, and $C$ the concentration of the state variable. Molecular diffusion coefficients were calculated using R-package marelac (Soetaert and Petzoldt, 2018). Bioturbation is depth-dependent, and assumes a constant biodiffusivity value $Db_0$ in a layer with

thickness $L_{mix}$. Below this depth, bioturbation decreases rapidly to zero, determined by the attenuation coefficient for bioturbation ($Db_{coeff}$).

$$Db_z = Db_0 e^{-(z - L_{mix})/Db_{coeff}} \qquad (Eq.\ 2)$$

### 2.1.2   Model parametrization

The model was parametrized for 5 different sedimentary settings in the Southern North Sea (Figure 1, Table 2): a coarse sandy

sediment (hereafter denoted as "Coarse") with a median grain size of 433 µm located on a sandbank in the Belgian part of the North Sea (BPNS). Two sediments with intermediate grainsizes (216 – 220 µm): one with a low nutrient load situated on the Dogger Bank ("FineL"), the other nearshore (BPNS) with a comparatively high nutrient load ("FineH"). Finally two muddy sediments with a high silt content (74 – 88 %): one with a low nutrient load situated offshore, on the Fladen Grounds ("MudL"), and one with a comparatively high nutrient load, situated nearshore in the BPNS ("MudH"). Biogeochemical data from these

stations were collected from boxcore samples in two separate sampling campaigns, one in September 2017 (Coarse, FineH, MudH), the other in May-June 2018 (FineL, MudL). For more details on sampling procedure see Toussaint et al. (2020), and De Borger et al. (2020)., respectively. Model parameters included both measured concentrations in the bottom water, as well as process rate parameters that were derived following the steady state fitting procedure as described in De Borger et al., (2020) (Table 4). In short, model parameters were determined by fitting the output of steady state OMEXDIA simulations to fluxes

of DIC and oxygen, and to measured porewater concentration profiles of $O_2$, $NH_4^+$, $NO_3^-$ and $PO_4^{3-}$, in a stepwise fitting procedure which combined manual parameter adjustment followed by an automated parameter optimization. The steady state model fits through measured solute profiles of $O_2$, $NO_3^-$, and $NH_4^+$ are shown in the supplemental information.

Using the steady-state condition as initial condition, several time variable boundary conditions were imposed for the dynamic model simulations. A sinusoidally varying carbon deposition flux with the model derived carbon flux (*Cflux*, Table

4) as the annual average, and imposing an amplitude of 1 was used as the upper boundary carbon flux (Figure 3A). Additional time variable boundary conditions (daily bottom water concentrations of $O_2$, $NH_4^+$, $NO_3^-$ and $PO_4^{3-}$, as well as bottom water temperature) were extracted from the Copernicus Marine Environmental Monitoring Service implementation of the ERSEM model (European Regional Seas Ecosystem Model, Butenschön et al., 2016; Copernicus Marine Service Information, 2020) for each location.



### 2.1.3 Disturbance modelling

Trawling disturbances were modelled as events causing the instantaneous removal of the surface layer due to hydraulic erosion (Depestele et al., 2016), followed by the mixing of a layer below that due to the actual gear penetration (Figure 2). The hydraulic erosion was implemented as a reset of the sediment water interface (SWI) to the depth of the eroded layer. The mixing was implemented as a homogenization of solids (FDET, SDET) over the mixing depth (Figure 2A), whereas solutes in the mixing depth ($O_2$, $NO_3^-$, $NH_4^+$, DIC) were set equal to the bottom water concentration of the respective solute to represent a complete flushing of the mixed layer with bottom water (Figure 2B).

Modelled trawling events also caused an immediate reduction in bioturbation rates, due to the mortality of benthic fauna after a trawl pass (Figure 3). Benthic mortality is mostly dependent on the total penetration depth of the gear (Hiddink et al., 2017), but also varies with habitat (Pitcher et al., 2017). The instantaneous reduction in bioturbation was included as a proportional depletion ($d$), dependent on the sediment type (lowest in coarse sand, highest in mud) and the penetration depth. It was calculated based on the total gear penetration depth ($TPD$, i.e. the sum of the eroded layer depth and the penetration depth, in cm), and the mud content (% $mud$) of the sediment, as described in Eq. 3 derived from Sciberras et al. (2018).

$$d = (TPD \cdot 3 + mud \times 0.3) / 100 \qquad \textit{Eq. 3}$$

The subsequent recovery of the bioturbation was modelled based on the logistic growth equation (Eq. 4), with the recovery rate ($r$) the inverse of the longevity of the species community, kept constant at 0.04 $y^{-1}$ (Rijnsdorp et al., 2016; Hiddink et al., 2019).

$$\frac{dN}{dt} = r \cdot N \left(1 - \frac{N}{K}\right) \qquad \textit{Eq. 4}$$

Here $N$ is the bioturbation rate ($cm^{-2}$ $d^{-1}$), with $K$ the full bioturbation rate for a species community at carrying capacity. The maximum reduction of the bioturbation was set to 90 %, to account for quasi immediate recolonization of trawled sediment by scavengers (Sciberras et al., 2018), and deeper living species that can survive intense trawling activity (Rijnsdorp et al., 2018).

## 2.2 Simulations

The model description was implemented in R (R Core Team, 2020), the concentration changes of simulated species due to transport were calculated using the R-package ReacTran (Soetaert and Meysman, 2012), and the resulting system of differential equations was solved using the deSolve package (Soetaert et al., 2010). Dynamic model simulations were initialized with a steady state solution calculated with annually averaged boundary conditions as input parameters, using the R-package rootSolve (Soetaert, 2009). This is necessary to build up an organic carbon, ammonium, ODU and DIC stock in the sediment (Soetaert et al., 1996b). Dynamic simulations were run for 15 years, with daily output, to generate sufficient independence from starting conditions. Reported modelling results stem from the last simulated year.

The frequency of the trawling events imposed ranged from 0 (the baseline) to 5 $y^{-1}$, based on realistic values of bottom trawling intensities in the North Sea (Rijnsdorp et al., 1998; Eigaard et al., 2017). Events were distributed randomly throughout the





year, given the absence of a clear seasonal pattern in trawling intensities (Rijnsdorp et al., 2020a). For each trawling frequency

and site, 30 model simulations were performed, each with a different pair of penetration and erosion depths, generated from a log-normal distribution of penetration depths. For the tickler chain trawl, a log normal distribution of penetration depths was generated given the average values (95% confidence limits) for sand and mud summarized in Pitcher et al. (2020): 3.2 (1.5, 6.7) cm and 1.9 (1.0, 3.7) cm for mud and sand respectively; the erosion depth was set to 22% of the penetration depth. The penetration depths for the pulse gear were set to 50 % of the tickler gear: 1.6 cm (0.75, 3.38) and 0.95 (0.49, 1.83) cm for mud

and sand. The erosion depth for the pulse gear was set to 70 % of the tickler gear (**Error! Reference source not found.**A; Depestele et al., 2016, 2019; Rijnsdorp et al., 2020b).

### 2.3 Statistical analysis

Linear models were constructed with selected model output variables to analyse the effects of the different gear types, trawling frequency, and the sedimentary context, on the rates of the different mineralization processes and on the total mineralization

(the sum of the separate mineralization processes). A normal distribution was adopted for the process responses. To deal with heterogeneity of variances of the residuals (for all models) in the linear models, a generalized least squares (GLS) structure was added (Pinheiro and Bates, 2000; Zuur et al., 2009; West et al., 2014), which allowed for unequal variances among treatment combinations to be included as a variance structure (Pinheiro and Bates, 2000; West et al., 2014). To find the most suitable variance structure, models with different variance structures were compared using AIC scores (Akaike, 1974), and

plots of fitted values and individual model terms versus the residuals (Zuur et al., 2009). For all models, a variance structure was selected that allowed for variances conditional on the station, and trawling frequency. Subsequently the fixed model component was optimized by manual stepwise selection, using the likelihood ratio test, and associated *p*-values as validation for removing excess terms (Zuur et al., 2009). During this step, the philosophy was adopted to not include significant interaction terms containing a certain variable, when said variable was not significant by itself. The minimal adequate model

was represented using restricted maximum likelihood estimation (REML, West et al., 2014). GLS models were implemented using the R package "nlme" (Pinheiro et al., 2019).

### 3 Results

### 3.1 Baseline model simulations

Baseline model simulations (undisturbed) show the differences in organic matter cycling between the chosen locations. In the

coarse sand station (Coarse), the average total mineralization rate was 13.6 mmol C m$^{-2}$ d$^{-1}$, with 89 % of this due to oxic mineralization, 6 % due to anoxic mineralization, and 5 % was denitrified (Table 5). The two muddy stations had either very high or very low total mineralization rates (MudH: 82 mmol C m$^{-2}$ d$^{-1}$, MudL: 8.5 mmol C m$^{-2}$ d$^{-1}$), and similar for the two fine sandy stations (FineH: 30 mmol C m$^{-2}$ d$^{-1}$, FineL: 8.1 mmol C m$^{-2}$ d$^{-1}$). Oxic mineralization dominated in FineL (oxic: 81 %, anoxic: 12 %, denit.: 6 %), FineH (oxic: 78 %, anoxic: 22 %, denit.: 0 %) and MudL (oxic: 72 %, anoxic: 18 %, denit.: 10



%), whereas the mineralization in the nearshore muddy station (MudH) was dominated by anoxic processes (oxic: 27 %, anoxic: 68 %, denit.: 5 %).

## 3.2 Impact on biota

Trawling-induced depletion of fauna substantially decreased average annual bioturbation rates. Bioturbation decreased with increasing penetration depth (Figure 3B, C), resulting in strongest decreases in muddy sediment (MudL, MudH), and larger
decreases in the deep penetrating gear versus the shallow penetrating gear (Table 3). In the Coarse sediment, the annually averaged bioturbation decreased gradually, from 81 % of its original value at 1 trawl $y^{-1}$ to 19 % at 5 trawls $y^{-1}$ in the tickler gear, and 49 % at 5 trawls $y^{-1}$ for the pulse gear. For the fine (FineL, FineH) and muddy (MudL, MudH) sediments the maximum depletion was reached after 4 (5 for the shallow gear) and 2 trawling events respectively.

## 3.3 Nutrient and organic carbon distribution

With higher trawling frequencies, concentrations of oxygen and nitrate in the sediment generally increased, whereas ammonium and organic carbon contents were always reduced (Figure 4). The magnitude of concentration changes was similar for both gear types (Table 5, mean percentage change relative to baseline concentrations reported). Increases in oxygen and nitrate concentrations were largest in the oligotrophic stations FineL and MudL, where concentrations of oxygen in the upper 5 cm increased 15 – 16 fold (resp. 1604 and 1516 %), while nitrate concentrations increased 9 - 19 fold (resp. 909 and 1911
%) at highest trawling intensities. In contrast, $O_2$ and $NO_3^-$ concentrations initially decreased at MudH by 25 and -50 % maximally for 1 – 2 trawls $y^{-1}$, before increasing by 52 to 81 % ($O_2$), and 123 to 188 % ($NO_3^-$) at 5 trawls $y^{-1}$. Ammonium ($NH_4^+$) concentrations decreased strongly in all sediments, with a decrease of up to 69 % in Coarse, 68 % in MudH, and > 90 % in FineL, FineH, MudL (Table 5).

Increasing the trawling frequency reduced the total amount of reactive organic carbon (labile + semi-labile, OC) in the
sediment, and reduced the penetration depth of the OC (Figure 5). Trawling frequencies of 3-5 $y^{-1}$ led to near-total depletion of reactive OC in all sediments in the upper 10 cm (> 95 % removed for Coarse and FineH, > 90 % for FineL, MudL and MudH, Table 5). The mean OC profiles for both gears at a given frequency were often visually different (dotted vs. full lines on Figure 5), but the average concentrations over 10 cm did not differ significantly. A redistribution of organic carbon was visible in the upper cm of the sediment, where organic carbon concentrations were higher in the impacted than in the baseline
simulation (example in the cutout of the top 5 mm shown for MudH, Figure 5Figure 5). In FineL, MudL and MudH the ratio of labile organic carbon (FDET) to semi-labile organic carbon (SDET) increased between 25 and 34 % (not shown). This effect was only noticeable in the upper 0.2 – 0.5 cm, below this depth values of this ratio in all trawling frequencies converged to 0, due to the depletion of labile organic carbon.



### 3.4 Total mineralization rates

The trawling frequency had a significant negative impact on all studied mineralization process rates (oxic, anoxic, denitrification), and on the total organic carbon mineralization, as confirmed by the negative coefficients in the GLS models ("Freq", Table 6**Error! Reference source not found.**). Changes in process rates also differed between the studied sediments, as seen by the inclusion of an interaction term between the sediment type and the trawling frequency (Freq:Stat). The sediment biogeochemical response to increasing trawling frequency was often nonlinear warranting the inclusion of a squared frequency

term ($Freq^2$). The gear type was only included as a significant explanatory variable in the model for denitrification, where the deeper penetrating gear (tickler) decreased denitrification rates more than the shallow penetrating gear (Table 6).

The total mineralization rate was impacted negatively in all cases, and decreases ranged from 5 % for 1 trawl $y^{-1}$ for MudL to -28.9 % for 5 trawls $y^{-1}$, for FineL (Table 7).

The change in oxic mineralization rates (base: 12.0, 23.3, 6.5, 6.1, 23.1 mmol $m^{-2}$ $d^{-1}$ for Coarse, FineL, FineH; MudL, and

MudH resp.) showed different patterns depending on the station (Table 7). For Coarse and FineH there was a consistent decrease in oxic mineralization rates with increasing trawling frequency, with maximum decreases at 5 trawls $y^{-1}$ of 21 % and 23 % for the tickler gear, and 21 % and 25 % for the pulse gear. In contrast, for FineL and MudL oxic mineralization increased at a frequency of 1 trawl $y^{-1}$ (8 % and 11 %), followed by a decrease at higher trawling frequencies with maximal decreases of 15 % and 10 % at 5 trawls $y^{-1}$ for the tickler gear; the values for the pulse gear were similar (Table 7). For MudH oxic

mineralization increased by a maximum of 56 % and 56 % at 5 trawls $y^{-1}$ for tickler and pulse gear respectively.

Anoxic mineralization rates (base: 0.9, 1.0, 6.6, 1.5, 57.7 mmol $m^{-2}$ $d^{-1}$ for Coarse, FineL, FineH; MudL, and MudH resp.) were also affected similarly by the two gear types and decreased for all stations, though with differing magnitudes. The lowest decrease was in the Coarse sediment, where the decrease in the anoxic mineralization rate was similar for all trawling frequencies (range of 18 % to 25 %), and the highest decrease was modelled at MudL, where anoxic rates decreased between

73 % and 83 %.

Denitrification rates (base: 0.6, 0.5, 0.1, 0.8, 4.2 mmol $m^{-2}$ $d^{-1}$ for Coarse, FineL, FineH; MudL, and MudH respectively) decreased with increasing trawling frequencies at 4 out of 5 stations, with a maximum reduction of 74 % (tickler) and 68 % (pulse) at FineL. Trawling frequencies of $1 - 2$ $y^{-1}$ caused increases in the denitrification for the Coarse sediment, and FineH and MudL. For the MudH trawls, denitrification rates increased by 50 and 49 % towards 5 trawls $y^{-1}$ for tickler and pulse gear

respectively.

As a result of the changes to denitrification, the removal of reactive N from the sediment changed accordingly. The sediments where denitrification decreased most (FineL, MudL), had 35 and 51 % of N produced by mineralization removed by denitrification when undisturbed and this reduced to 11 % and 45 % respectively for 5 trawls $y^{-1}$. For the Coarse sediment the fraction of N removed increased from 26 % when undisturbed to 30 % for 1 trawl $y^{-1}$, and then decreased again to 25 %. In

MudH more N was removed as well, with a near-doubling as a peak at 5 trawls $y^{-1}$ (48 %, up from 26 % as the base).





All previous results represent average changes throughout the year, but trawling also showed instantaneous effects, as illustrated by the decrease of denitrification rates (to nearly 0 mmol m$^{-2}$ d$^{-1}$) immediately after a trawl events (Figure 6).

### 3.5 Relative changes

The relative contribution of the mineralization processes to the total mineralization changed markedly between trawling
frequencies and stations (Figure 7). In general, the proportion of oxic mineralization increased, at the expense of anoxic mineralization. The largest changes occurred when switching from 0 to 1 trawling event y$^{-1}$, and values remained stable from 2 events y$^{-1}$ onwards. The proportion of oxic mineralization increased most at MudH (tickler: 116 %, pulse: 112% for 5 trawls y$^{-1}$), and smallest changes occurred at Coarse (<1% for both gears). The proportion of anoxic mineralization on the other hand, decreased in all simulations (Figure 7). Largest changes were modelled at FineL and MudL (69 % and 78 % respectively), and
smallest for Coarse and FineH (14 and 18 % respectively). The proportion of mineralization performed by denitrification decreased in FineL and MudL (71 % and 10 %, 5 events y$^{-1}$), doubled at MudH (100 %) and remained practically the same for Coarse and FineH.

### 4 Discussion

#### 4.1 Organic carbon depletion

Simulated trawling of the seafloor impacted the sediment biogeochemistry in all environments, and for all trawling frequencies. The amount of total mineralizable carbon in the sediment consistently decreased with higher trawl frequencies, but the changes in mineralization pathways differed from case to case. The main drivers of the biogeochemical changes were found to be the depletion of organic carbon (OC) in the sediment (i.e. the substrate for mineralization itself), the redistribution of this OC nearer to the SWI (Figure 5), and the increasing oxygenation of the sediment. With each trawl pass, a part of the organic carbon
is removed along with the top sediment layer, preventing the OC to be transported to deeper layers through advective or bioturbation processes (Figure 5). In addition, part of the benthos in the sediment is removed, often strongly decreasing the bioturbation rate (especially after multiple trawling events y$^{-1}$, Table 3), which also affects the rate at which organic matter is distributed in the sediment. Thus, with increasing trawling frequency, organic matter is concentrated more near the sediment water interface, and becomes more vulnerable to erosion in subsequent trawling events. The fine sandy station with low organic
matter content (FineL), as well as both muddy stations (MudL, mudH), showed smaller decreases in surface organic carbon concentrations compared to the eutrophic fine sandy (FineH) and Coarse sediments (Table 5), mainly because baseline bioturbation rates in the former were 3 orders of magnitude larger (Table 4). As such, bioturbation seems to cause an increased resistance to carbon loss by facilitating transport to deeper layers, making it less vulnerable to surface disturbances.

While the physical OC depletion caused by the penetrating gear is aggravated by the loss of bioturbating fauna in the
sediment, this effect is context-dependent as bioturbators show variable levels of resistance to trawling (Hale et al., 2017; Tiano et al., 2020). Our modelled results provide further evidence that surviving fauna help buffer and mitigate the



biogeochemical effects of trawling (Duplisea et al., 2001). Tiano et al. (2019) observed decreases in sedimentary chl *a* in the upper 1 cm immediately after trawling, of 41 and 83 % for pulse and tickler gears respectively. Also, OM depletion as a result of long term fishing has been reported, even at water depths beyond 500 m (Martín et al., 2014; Paradis et al., 2019), where

comparisons between trawled and untrawled sites yielded a difference in OC between 20 and 60 % (Paradis et al., 2019). Our results contrast with studies that found enhanced OC concentrations in trawled areas (Palanques et al., 2014; Pusceddu et al., 2005; Sciberras et al., 2016), possibly due to differing hydrodynamic and morphological conditions of the North Sea compared to other areas.

The decrease in total mineralization rates may partly be offset by re-deposition of organic matter, which was not

considered in our model. Not all eroded organic matter stays in the water column, but a part resettles on the sediments. How this redistribution occurs depends on the sediment type and the local hydrodynamics, which determine the distance over which eroded sediment particles are transported (Le Bot et al., 2010; Robinson et al., 2005). It can be expected that for coarser, heavier sediments, a fraction will be redeposited in the trawling track, but that for muddy sediments, lighter and rich in organic matter (Mayer, 1994), the majority of eroded material remains in suspension long enough to be transported elsewhere. In the

North Sea, suspended material is transported from the Southern Bight northward by counterclockwise residual currents. Ultimately, materials are deposited in the Skagerrak (Dauwe et al., 1998). So given the intensity, and the persistence with which vast areas of the southern North Sea are trawled (e.g. total annual sediment mobilization by the Dutch trawling fleet varied between 8 and 17 $10^{14}$ kg of sediment between 2010 and 2016, Rijnsdorp et al., 2020a), we expect that trawling-induced sediment resuspension plays a significant role in the northward transport, and actively contributes to organic matter depletion

in southern areas. Additionally, Van De Velde et al. (2018) found an increase in mineralization rates of over 200 % after a disturbance event in muddy sediments (from the same origin as MudH). This was attributed to multiple possible factors, such as 'self-priming' by mixing refractory with labile organic matter, burial of phytoplankton in settling sediment, and the introduction of oxygen to redox shuttle mechanisms. Our results show a trawl-induced enhancement of total oxic mineralization at mudH, but no increase in the total mineralization rate, perhaps because the aforementioned processes were

not included in the used model.

### 4.2 Changes to mineralization pathways

Trawling activities generally caused strong increases in sedimentary oxygen and nitrate availability, and decreases in the ammonium content (Table 5, Figure 4). Sediment oxygenation increased both because of a direct injection of oxygen rich bottom water in deeper sediment layers during a trawling event, and because oxygen consumption by mineralization processes

decreased as a result of strong decreases in OC and ammonium. As a result of increased oxygen availability, the importance of oxic mineralization generally increased with trawling, whereas anoxic mineralization decreased (Figure 7).

Strongest increases in the proportion of oxic mineralization were modelled for the sediment characterized by a high silt percentage and organic matter load (mudH). These types of sediments also have a low permeability, high mineralization rates and a low oxygen penetration depth (Braeckman et al., 2014), with a lesser importance of oxic relative to anoxic mineralization



in undisturbed conditions. Fishing gears penetrate deepest in these muds, and as such provide oxygen to deeper layers, although this is consumed rapidly.

The higher oxygen concentrations also had a clear inhibiting effect on denitrification rates (Figure 6). Denitrification in coastal shelf sediments accounts for an estimated third of all nitrogen loss in Earth's marine surfaces (Middelburg et al., 1996), making these regions crucial to counteract nitrogen eutrophication (Galloway et al., 2004; Seitzinger et al., 2006). For all stations,

trawling events caused an instantaneous dip in denitrification rates, because of the injection of $O_2$, and removal of the electron acceptor ($NO_3^-$) from the sediment, similar to the results of van der Molen et al. (2013) (Figure 6, black vertical lines). However, a discrepancy was noted between the biogeochemical impacts of trawling in cohesive sediments with high organic matter concentrations versus sandier and comparatively low nutrient sediments, consistent with literature findings (Polymenakou et al., 2005; Van De Velde et al., 2018; Tiano et al., 2019). The coarsest sandy sediments (Coarse) were by default deeply

oxygenated (Figure 4 A, B), with denitrification maximally inhibited by oxygen concentrations. The increasing trawling frequency had little effect on oxygen penetration, and nitrate concentrations only marginally increased (Figure 6A, B), resulting in minor changes to mineralization pathways such as denitrification on average, although instantaneous effects could be prominent (Figure 6 C). In oligotrophic finer sediments (FineL, mudL), there was a massive increase in both oxygen and nitrate concentrations as a result of trawling (Figure 4 D, E; figure 6 D, E). Whereas increasing $NO_3^-$ concentrations would stimulate

denitrification on their own, the rise of oxygen concentrations strongly inhibited denitrification, leading to a drop in denitrification rates throughout the year (Figure 6 F). The more eutrophic fine sediment FineH displayed a similar pattern of increased oxygenation mineralization as the other fine sandy sites, but the already low baseline denitrification rates could not decrease further. In the unperturbed simulation of MudH, mineralization was predominantly anoxic (68 %), with denitrification limited by $NO_3^-$ availability. Increasing the oxygenation in this type of sediment caused the denitrification to double, by

increasing the nitrate availability (Figure 6 G-I).

Ferguson et al. (2020) found that denitrification rates in Moreton Bay, Australia were reduced between 11 and 50 %, within 3 hours after a trawling event, and this rate decreased after successive trawling events during the studied period. These decreases were attributed to homogenization of the sediment, which removes oxic microniches created by fauna, and thus zones of intense coupled nitrification-denitrification (Ferguson et al., 2020). Though the key role of redox microniches is not directly

investigated here, we acquired decreases in denitrification rates in a similar range.

Within the marine environment, sediments are sites characterized by high nutrient concentrations, therefore offering resilience against reductions in nutrient loadings. Soetaert and Middelburg (2009) showed that storage of ammonium in sediments significantly delays the response of shallow systems to oligotrophication, as the efflux of nitrogen from the sediment will compensate part of the losses in the water column. The increased reduction of the ammonium concentrations with

fishing intensity will affect this buffering capacity of the sediment. The nitrogen buffering capacity of the investigated sediments, representative for a large fraction of North Sea sediments, was affected similarly by both gears. Firstly, the stock of nitrogen (as $NH_4^+$) in the sediment was directly affected by porewater flushing during trawling (with decreases > 99 % in



some cases, Figure 4 F). Secondly, lower availability of reaction substrate (OC, $NH_4^+$) decreased denitrification rates, reducing N removal to the atmosphere.

### 4.3 Reducing gear penetration depth

Though there were minor differences in impacts to mineralization between the deep and shallow penetrating gears (e.g. 3.2 ± 1.2 cm vs. 1.6 ± 0.6 cm in mud), the fishing gear type was only included as a significant predictor for the denitrification rates, with a small coefficient (-0.00038; Table 6). This is because the freshly deposited stocks of organic carbon are present near the sediment surface, and any gear that penetrates the sediment impacts this layer, especially for multiple trawling events per year (Figure 5, Table 5). This suggests that only a thin layer of surface sediment needs to be disturbed to generate significant biogeochemical changes (Dounas et al., 2005).

For pulse trawling (PulseWing) it has been observed that the layer of oxidized sediments recovered from disturbed to steady state conditions within 48 hours, while this recovery for the deeper penetrating tickler (SumWing) trawling exceeded this time (Depestele et al., 2019). Tiano et al. (2019) found an increase in oxygen penetration depths after trawling events with both gears, associated to a decrease in surface chl *a* and benthic mineralization, while oxygen consumption in the water column temporarily increased due to resuspended sedimentary material. Many biogeochemical processes are mediated by the dynamics of oxygen near the SWI, which itself is influenced by the composition, and permeability of the sediment.

A shift towards fining (an increased proportion of finer grain size classes) has been described in certain trawled areas, with expected consequences for sediment biogeochemistry, such as an increased rate of sulphate reduction (Trimmer et al., 2005). But the opposite occurs just as well (Depestele et al., 2019; Mengual et al., 2019; Tiano et al., 2019). In these cases resuspended silt is exported away from the trawling site, leaving a coarsened trawling track, with the results subtly different between gear types (Depestele et al., 2019; Tiano et al., 2019). This means that the effects of fishing gears on grain size sorting should be better characterized for various sediment types, to constrain the uncertainty around predictions of gear impacts on sediment.

Other studies have reported clear positive effects of reducing the penetrations depths of fishing gears, such as decreased sediment mobilization and homogenization, and reduced organic matter depletion (Depestele et al., 2019; Tiano et al., 2019). In our work the differences with respect to organic matter mineralization dynamics between deep and shallow penetrating gear types mostly remained suggestive, rather than statistically conclusive. Additionally, largest impacts occurred when increasing the trawling frequency from 0 to 1 trawling event per year, and the response of mineralization processes to increased trawling frequency was often non-linear, making them more difficult to predict. This would imply that management strategies aimed at maintaining the ecosystem functions provided by shelf sediments should be focused on spatial controls, bottom impact quotas, and effort control, rather than on technical improvements to the trawling gear that per definition require contact with the bottom to catch commercially viable target species (McConnaughey et al., 2020). An effective strategy limits the impacted surface area, and allows carbon stocks and faunal communities in the sediment to recover from a disturbance, resulting in the recovery of vital biogeochemical functions such as denitrification and carbon burial.

Shifting the fishing effort from peripheral areas to core fishing grounds would, for example, reduce the area where the top sediment layer is removed on a regular basis, along with the associated reduction in mineralization of organic matter. Additionally, this can be achieved through time-restricted bottom trawling, in which fishing grounds are closed off temporarily. For this, the site-specific conditions such as rates of biogeochemical recovery and sedimentation rates would need to be
investigated to fine-tune management (Paradis et al., 2019). Alternatively, areas with high denitrification rates, crucial for eutrophication mitigation can be closed to trawling completely, as done in other regions of the world (Ferguson et al., 2020).

## 5.    Conclusion

With the addition of perturbation events to a model of early diagenesis, and a description of faunal mortality and recovery, we simulated the effects of increasing bottom trawling frequencies on sediment biogeochemistry. The results showed that bottom
trawl fisheries strongly impacted the sediment biogeochemistry, and the magnitudes of the changes were dependent on the sedimentary context and trawling frequency. Two types of fishing gears were investigated. The exposed top sediment layer rich in organic matter was targeted similarly by both fishing gears, resulting in a similar loss of organic carbon, which was further exacerbated by the loss of bioturbating fauna. A shift towards increasingly oxic mineralization at the cost of anoxic mineralization was observed, driven by an often strongly increased oxygen availability in the sediment. The removal of fixed
nitrogen by denitrification was not affected similarly in all sediments. Denitrification increased in nearshore cohesive mud, and decreased elsewhere, with highest decreases in offshore sediments with lower carbon loads. Our modelling results corroborate multiple patterns found in other studies, and can serve to interpret research and search for mitigation strategies. Trawling impacts are hard to mitigate by only reducing the penetration depth of the gear, so additional management strategies are needed to allow for partial, or full recovery of biogeochemical functions in-between trawling events.

**Code availability**

Model code and instructions for producing similar output will be made available on request to the corresponding author.

**Author contribution**

EDB, JT, KS, and AJR devised the study, and contributed to the manuscript. UB contributed to the manuscript. EDB collected field data used for the model descriptions and performed the model simulations. KS developed the dynamic modelling
environment in R.

**Competing interests**

None of the authors have any financial benefits or other conflicts of interest resulting from the publication of this manuscript.



**Acknowledgements**

E.D.B. is a doctoral research fellow funded by the Belgian Science Policy Office BELSPO, contract BR/154/A1/FaCE-It. J.T.
is a doctoral research fellow funded by the European Maritime and Fisheries Fund EMFF, and the Netherlands Ministry of
Agriculture Nature and Food Quality LNV (Grant/Award Number: 1300021172). U.B. is a postdoctoral research fellow at
Research Foundation - Flanders (FWO, Belgium) (Grant 1201720N). We thank Toussaint et al. for the data needed to
parametrize the nearshore sediments used in this modelling exercise.

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





**Figures**

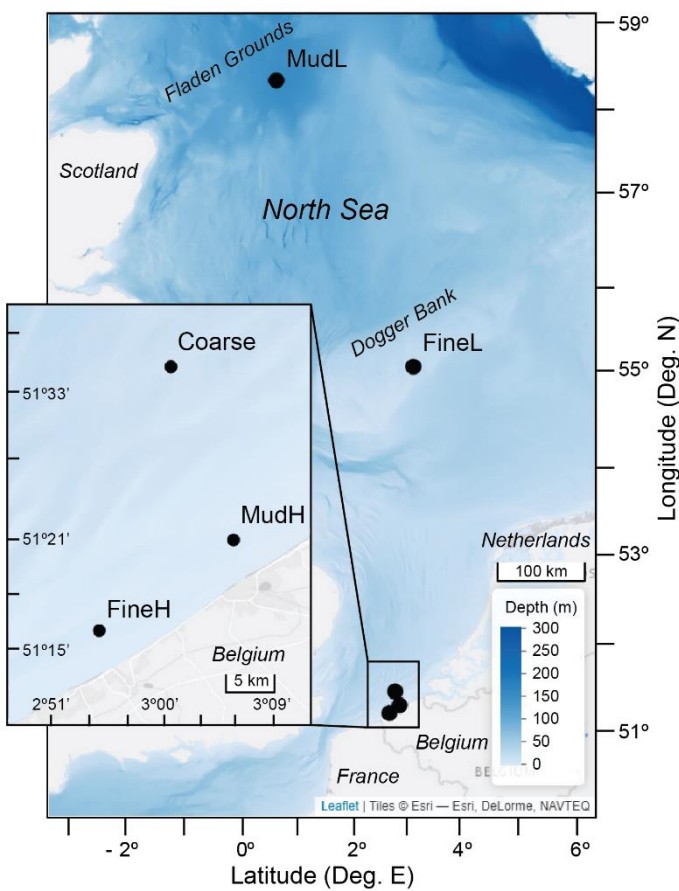

**Figure 1: Sampling locations in the North Sea. Offshore stations MudL and FineL sampled by De Borger et al. (2020), and nearshore stations (inset) in the Belgian part of the North Sea by Toussaint et al. (2020). L = low nutrient content relative to H high nutrient content. Basemap: © Esri, depth raster by GEBCO Compilation Group (2020).**




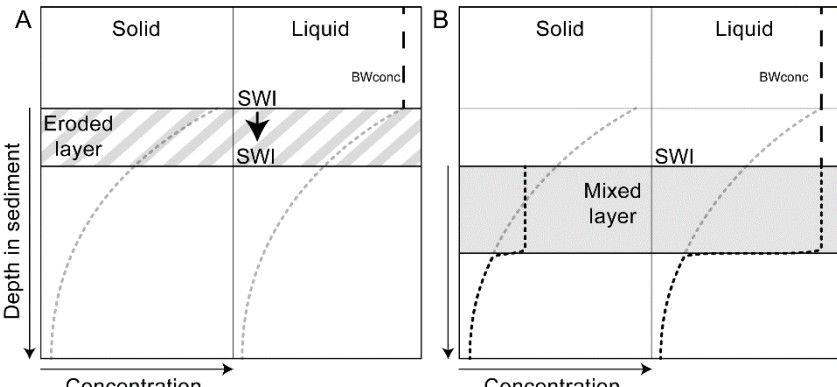

**Figure 2: Implementation of a trawling event, on sediment concentrations of solids and liquids. Dotted grey lines: initial concentration profile, black line: profile after the event. (A) Hydraulic erosion removes a layer of sediment, moving the sediment-water interface (SWI) downward; the effect is implemented similarly for solids and liquids. (B) Subsequent mixing of the sediment homogenises the solid concentration over the mixed layer depth whereas liquids are set to the overlying bottom water concentration (BWconc). Depths of both impacts are defined in the text.**


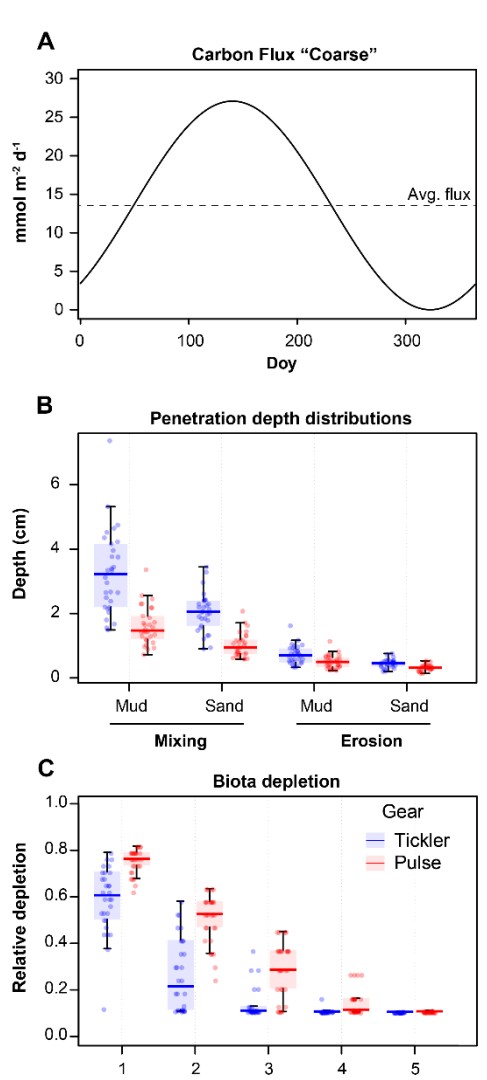

**Figure 3: (A) Example of sinusoidal varying carbon flux, based on average DIC efflux measured by Toussaint et al. (2020) for the**
**Coarse sediment. (B) Imposed mixing and erosion depths (cm), for fine sandy and muddy sediments, for the two gear types (tickler chain and pulse gear). (C) Simulated depletion of bioturbation, relative to the maximum (y-axis), with increasing trawling frequency (y$^{-1}$, x-axis) in fine sands. Doy = day of year.**





**Figure 4: Range of nutrient concentrations (gray lines) throughout the year, and average annual concentration (black line) in an untrawled sediment, and sediment trawled 3 y$^{-1}$ of Coarse (A-C), FineL (D-F), and MudH (G-I). Nutrient concentrations in mmol m$^{-3}$. Note the difference in depth range (y-axis) shown for the different sediments.**

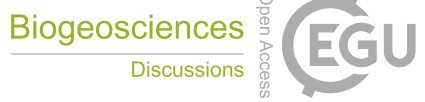

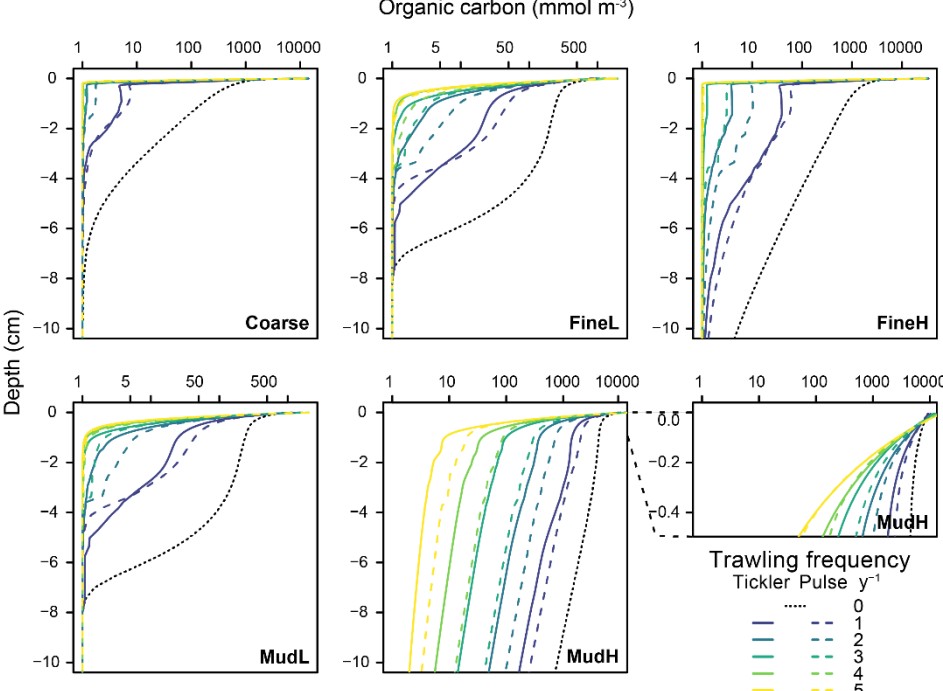

**Figure 5: Total organic carbon concentrations in the upper 10 cm of the sediment (mmol m$^{-3}$) for the different stations (X-axis on log-scale). Profiles represent the average profile per trawling intensity ($n$ = 30), for the deep penetrating gear (full), and shallow penetrating gear (dotted line). For station MudH a cut-out of the first 0.5 cm is inflated to show the displacement of OC to the surface.**



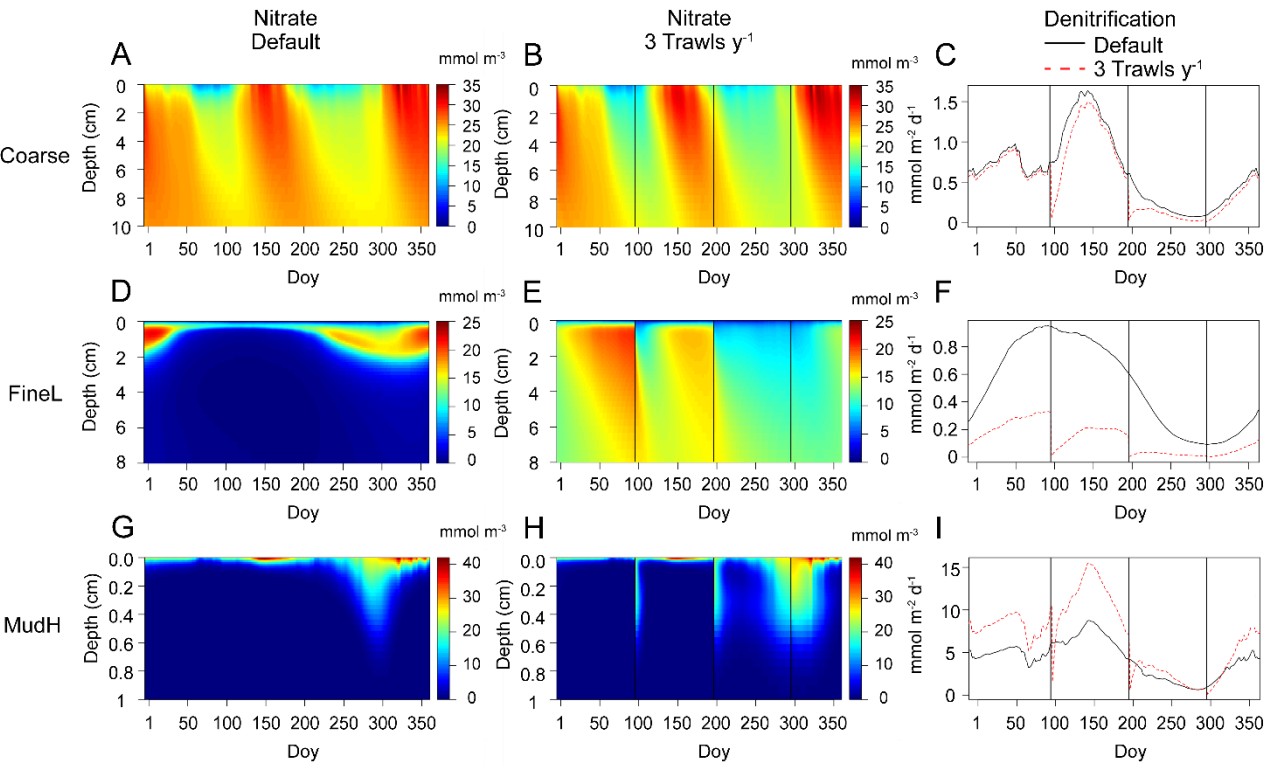

**Figure 6: Nitrate concentrations (mmol m$^{-3}$) in the sediment for a no-trawling simulation (A, D, G), and for 3 trawls y$^{-1}$ (D, E, F), and associated annual denitrification rates (mmol m$^{-2}$ d$^{-1}$, black lines = default (0 trawls y$^{-1}$), red dotted lines = trawling simulation). Represented stations are Coarse (A – C), FineL (D - F), and MudH (G - I). Black vertical lines indicate when trawling events take place.**





**Figure 7: Relative contributions (x-axis, as proportion) of the three main mineralization processes (left column: oxic; middle: anoxic; right: denitrification) to the total mineralization for each gear type (blue boxes: tickler gear; red boxes: pulse gear), and for increasing trawling frequency (y-axis, y⁻¹).**









## Tables

**Table 1: Diagenetic reactions used in OMEXDIA. x denotes the molar C:P ratio, y the molar N:P ratio in organic matter per mole of phosphorus (for Redfield Stoichiometry, x = 106, y = 16).**

| Process | Reaction | |
|---------|----------|---|
| Oxic mineralization | $(CH_2O)x\ (NH_3)y\ (H_3PO_4) + xO_2 \rightarrow xCO_2 + yNH_3 + H_3PO_4 + xH_2O$ | (1) |
| Denitrification | $(CH_2O)x\ (NH_3)y\ (H_3PO_4) + 0.8 \cdot HNO_3 \rightarrow xCO_2 + yNH_3 + 0.4 \cdot N_2 + H_3PO_4 + 1.4 \cdot H_2O$ | (2) |
| Anoxic mineralization | $(CH_2O)x\ (NH_3)y\ (H_3PO_4) + an\ oxidant \rightarrow xCO_2 + yNH_3 + H_3PO_4 + xODU + xH_2O$ | (3) |
| Nitrification | $NH_3 + 2O_2 \rightarrow HNO_3 + H_2O$ | (4) |
| ODU oxidation | $ODU + O_2 \rightarrow an\ oxidant$ | (5) |

**Table 2: Characteristics of the selected sites. Low and High nutrient classification is based on relative differences in nutrient build-up for the same sediment type (see figure S1 in supplement).**

| Sediment type | Nutrients | Name | Lat (Deg. N) | Lon (Deg. E) | Depth (m) | MGS (µm) | SD.9 (µm) | SD.1 (µm) | VFines (%) | Silt (%) | Source* |
|---------------|-----------|------|-----|-----|-------|-----|------|------|--------|------|---------|
| Coarse sand | Low | Coarse | 51.43483 | 2.809822 | 22 | 433 ± 43 | 660 ± 67 | 286 ± 31 | 0 ± 0 | 0 ± 0 | a |
| Fine sand | Low | FineL | 55.17374 | 3.161264 | 26 | 216 ± 2 | 328 ± 7 | 143 ± 0 | 3 ± 0 | 0 ± 0 | b |
| | High | FineH | 51.1853 | 2.7013 | 9 | 220 ± 8 | 394 ± 67 | 91 ± 61 | 4 ± 2 | 9 ± 5 | a |
| Mud | Low | MudL | 58.20097 | 0.525871 | 148 | 24 ± 1 | 67 ± 3 | 4 ± 0 | 10 ± 0 | 88 ± 1 | b |
| | High | MudH | 51.2714 | 2.905033 | 11 | 19 ± 1 | 208 ± 28 | 3 ± 0 | 4 ± 1 | 74 ± 5 | a |

**Sources: (a) Toussaint et al. (2020), (b) De Borger et al. (2020)..**


**Table 3: Percentage of bioturbation (average ± sd) remaining after sustained trawling activity at a given trawling frequency ($y^{-1}$), for the different sediment types. T: deeply penetrating tickler gear, P: shallow penetrating pulse gear.**

| Frequency | Coarse | | Fine | | Mud | |
|-----------|--------|--------|--------|--------|--------|--------|
| | T | P | T | P | T | P |
| 1 | 81 ± 7 | 90 ± 3 | 60 ± 16 | 76 ± 7 | 27 ± 17 | 42 ± 12 |
| 2 | 61 ± 13 | 80 ± 14 | 26 ± 16 | 51 ± 1 | 11 ± 1 | 11 ± 1 |
| 3 | 42 ± 17 | 69 ± 21 | 14 ± 7 | 28 ± 12 | 11 ± 0 | 11 ± 0 |
| 4 | 27 ± 18 | 59 ± 25 | 11 ± 1 | 15 ± 6 | 11 ± 0 | 11 ± 0 |
| 5 | 19 ± 15 | 49 ± 26 | 11 ± 1 | 11 ± 1 | 11 ± 0 | 11 ± 0 |





**Table 4: Parameters used for modelling the different sediment types.**

| Parameter | Description | Unit | Coarse | FineL | FineH | MudL | MudH |
|---|---|---|---|---|---|---|---|
| **a.** | **Fitted parameters** | | | | | | |
| **wSed** | Advection rate | cm d$^{-1}$ | $3.00\ 10^{-4}$ | $3.00\ 10^{-7}$ | $3.00\ 10^{-4}$ | $3.00\ 10^{-7}$ | $3.00\ 10^{-4}$ |
| **pFast** | Fast degrading fraction organic matter | - | 0.94 | 0.95 | 0.93 | 0.95 | 0.90 |
| **pSlow** | Slow degrading fraction organic matter | - | 0.06 | 0.05 | 0.07 | 0.05 | 0.10 |
| **rFast** | Decay rate FDET | d$^{-1}$ | 0.07 | 0.05 | 0.05 | 0.05 | 0.06 |
| **rSlow** | Decay rate SDET | d$^{-1}$ | $3.82\ 10^{-4}$ | $1.00\ 10^{-4}$ | $1.86\ 10^{-4}$ | $1.08\ 10^{-4}$ | $3.65\ 10^{-5}$ |
| **Db** | Biodiffusivity coefficient | cm$^2$ d$^{-1}$ | $0.10\ 10^{-6}$ | $1.03\ 10^{-3}$ | $2.18\ 10^{-6}$ | $1.11\ 10^{-3}$ | $1.73\ 10^{-3}$ |
| **biotdepth** | Mixed layer depth | cm | 0.50 | 2.00 | 2.00 | 2.00 | 0.50 |
| **rnit** | Max. nitrification rate | d$^{-1}$ | 1.76 | 24.84 | 1.92 | 11.62 | 3.62 |
| **rODUox** | Max. ODU oxidation rate | d$^{-1}$ | 0.70 | 3.82 | 2.26 | 5.92 | 3.58 |
| **ksO2oduox** | Half saturation, $O_2$ in ODU oxidation | mmol $O_2$ m$^{-3}$ | 1.11 | 2.74 | 0.69 | 4.61 | 3.50 |
| **ksNO3denit** | Half saturation, $NO_3^-$ in denitrification | mmol $NO_3$ m$^{-3}$ | 72.73 | 48.02 | 68.70 | 14.47 | 8.80 |
| **kinO2denit** | Half saturation, $O_2$ inhibition of denitrification | mmol $O_2$ m$^{-3}$ | 82.63 | 57.69 | 33.56 | 86.55 | 77.00 |
| **kinNO3anox** | Half saturation, $NO_3^-$ inhibition anoxic mineralization | mmol $NO_3$ m$^{-3}$ | 6.44 | 1.26 | 6.30 | 2.33 | 8.67 |
| **kinO2anox** | Half saturation, $O_2$ inhibition anoxic mineralization | mmol $O_2$ m$^{-3}$ | 91.00 | 57.69 | 95.08 | 42.19 | 75.84 |
| **rCaPprod** | Rate of CaP production | d$^{-1}$ | 0.08 | $1.85\ 10^{-3}$ | $2.19\ 10^{-3}$ | 0.00 | $9.82\ 10^{-7}$ |
| **rCaPdiss** | Rate of CaP dissolution | d$^{-1}$ | $6.60\ 10^{-5}$ | $9.01\ 10^{-9}$ | 0.00 | 0.00 | $4.47\ 10^{-6}$ |
| **rFePadsorp** | Rate of FeP adsorption | d$^{-1}$ | 0.094 | 0.16 | 4.06 | 0.10 | 0.36 |
| **rFePdesorp** | Rate of FeP desorption | d$^{-1}$ | 0.00 | $9.26\ 10^{-5}$ | 0.02 | 0.00 | 0.11 |
| **b.** | **Measured parameters** | | | | | | |
| **Cflux** | Carbon deposition flux | nmolC cm$^{-2}$ d$^{-1}$ | 1354.7 | 810.9 | 2994.0 | 848.0 | 9025.5 |
| **Φ** | Porosity | - | 0.35 | 0.59 | 0.42 | 0.71 | 0.73 |





**Table 5: Overview of effect of increasing trawling intensities for tickler gears, and pulse gears (columns) on sedimentary concentrations of O₂, NO₃⁻, NH₄⁺ (top 5 cm) and organic carbon (top 10 cm). The baseline scenario (frequency = 0) is displayed as the absolute concentration (mmol m⁻³), increasing frequencies (1 – 5 y⁻¹) are shown as % change of the baseline rate (+ = increase, - = decrease). Concentrations are the average annual concentrations, in mmol m⁻³ for solutes, and mol m⁻³ for organic carbon.**


| | | | Tickler | | | | | | Pulse | | | | |
|---|---|---|---|---|---|---|---|---|---|---|---|---|---|
| | | Conc. | --- % Change --- | | | | | Conc. | --- % Change --- | | | | |
| | station | 0 | 1 | 2 | 3 | 4 | 5 | 0 | 1 | 2 | 3 | 4 | 5 |
| **O₂** *mmol m⁻³* | Coarse | 117.1 | 96 ± 7 | 97 ± 4 | 98 ± 3 | 99 ± 3 | 99 ± 4 | 117.1 | 98 ± 8 | 98 ± 3 | 100 ± 3 | 99 ± 4 | 97 ± 2 |
| | FineL | 8.1 | 239 ± 169 | 805 ± 280 | 1211 ± 387 | 1637 ± 270 | 1604 ± 347 | 8.1 | 133 ± 136 | 447 ± 195 | 1013 ± 300 | 1335 ± 364 | 1584 ± 323 |
| | FineH | 9.3 | 656 ± 142 | 1081 ± 79 | 1075 ± 165 | 1036 ± 254 | 965 ± 253 | 9.3 | 686 ± 322 | 1053 ± 202 | 1153 ± 158 | 1196 ± 217 | 1129 ± 212 |
| | MudL | 8.5 | 467.4 ± 182 | 1010 ± 208 | 1296 ± 272 | 1512 ± 311 | 1517 ± 287 | 8.5 | 296 ± 154 | 891 ± 169 | 1340 ± 392 | 1578 ± 351 | 1619 ± 264 |
| | MudH | 1.8 | -28.8 ± 4 | -25 ± 38 | 14 ± 57 | 27 ± 53 | 52 ± 57 | 1.8 | -25 ± 4 | -27 ± 21 | -9 ± 44 | 52 ± 61 | 81 ± 49 |
| **NO₃⁻** *mmol m⁻³* | Coarse | 23.5 | 11.3 ± 7 | 10 ± 8 | 6 ± 9 | 1 ± 9 | 0 ± 8 | 23.5 | 13 ± 7 | 9 ± 9 | 6 ± 10 | 4 ± 9 | -4 ± 4 |
| | FineL | 2.2 | 991 ± 135 | 1038 ± 52 | 967 ± 90 | 882 ± 87 | 909 ± 129 | 2.2 | 866 ± 251 | 1044 ± 44 | 981 ± 53 | 932 ± 87 | 882 ± 105 |
| | FineH | 2 | 584 ± 88 | 690 ± 100 | 655 ± 108 | 585 ± 132 | 565 ± 132 | 2 | 559 ± 202 | 685 ± 115 | 690 ± 89 | 675 ± 79 | 626 ± 89 |
| | MudL | 1.2 | 1598 ± 114 | 1733 ± 81 | 1779 ± 128 | 1822 ± 246 | 1912 ± 253 | 1.2 | 1464 ± 312 | 1687 ± 60 | 1662 ± 119 | 1675 ± 167 | 1741 ± 187 |
| | MudH | 0.2 | -51 ± 10 | -40 ± 61 | 39 ± 123 | 70 ± 117 | 123 ± 125 | 0.2 | -50 ± 11 | -47 ± 31 | -12 ± 85 | 114 ± 136 | 188 ± 104 |
| **NH₄⁺** *mmol m⁻³* | Coarse | 0.9 | -53 ± 2 | -57 ± 3 | -59 ± 2 | -62 ± 2 | -63 ± 2 | 0.9 | -52 ± 4 | -57 ± 3 | -59 ± 2 | -62 ± 2 | -64 ± 2 |
| | FineL | 41 | -97 ± 2 | -99 ± 0 | -99 ± 0 | -100 ± 0 | -99 ± 1 | 41 | -94 ± 6 | -99 ± 1 | -99 ± 0 | -99 ± 0 | -100 ± 0 |
| | FineH | 125.6 | -98 ± 1 | -99 ± 0 | -99 ± 1 | -98 ± 3 | -97 ± 6 | 125.6 | -96 ± 6 | -99 ± 1 | -100 ± 0 | -99 ± 1 | -99 ± 2 |
| | MudL | 50.7 | -98 ± 1 | -99 ± 0 | -99 ± 0 | -99 ± 1 | -99 ± 0 | 50.7 | -97 ± 3 | -99 ± 0 | -99 ± 0 | -99 ± 0 | -99 ± 0 |
| | MudH | 21612 | -50 ± 7 | -62 ± 3 | -65 ± 6 | -63 ± 6 | -61 ± 9 | 2162 | -43 ± 8 | -59 ± 5 | -64 ± 4 | -68 ± 6 | -68 ± 7 |
| **Organic C** *mol m⁻³* | Coarse | 78.4 | -87 ± 2 | -92 ± 2 | -94 ± 1 | -95 ± 1 | -96 ± 1 | 78.4 | -86 ± 2 | -92 ± 2 | -94 ± 2 | -95 ± 1 | -96 ± 1 |
| | FineL | 94.9 | -72 ± 7 | -84 ± 3 | -87 ± 3 | -90 ± 2 | -91 ± 1 | 94.9 | -65 ± 12 | -81 ± 6 | -86 ± 2 | -89 ± 2 | -91 ± 1 |
| | FineH | 326.3 | -86 ± 3 | -93 ± 2 | -95 ± 1 | -96 ± 1 | -96 ± 1 | 326.3 | -84 ± 4 | -93 ± 2 | -94 ± 2 | -96 ± 1 | -96 ± 1 |
| | MudL | 92.4 | -71 ± 8 | -84 ± 4 | -87 ± 2 | -89 ± 2 | -90 ± 2 | 92.4 | -65 ± 12 | -84 ± 5 | -87 ± 3 | -89 ± 2 | -91 ± 1 |
| | MudH | 2456 | -70 ± 11 | -82 ± 6 | -90 ± 3 | -90 ± 1 | -91 ± 0 | 2456 | -63 ± 15 | -75 ± 11 | -83 ± 6 | -85 ± 3 | -90 ± 1 |





**Table 6: Generalized least squares (GLS) models for the total mineralization, oxic mineralization; anoxic mineralization, and denitrification as a function of increasing trawling frequency (Freq and Freq$^2$), the fishing gear type (Gear), and the sediment context (Sed) and interactions between model terms Freq:Stat, Freq$^2$:Stat.**

| Response variable y | Model |
| --- | --- |
| **Total mineralization** | y = Intercept + a · Freq + b · Freq$^2$ + c · Stat + d · Freq:Sed + e · Freq$^2$:Sed |
| | y = 29.96 − 1.93 Freq + 0.07 Freq$^2$ + 55.05 S1* − 21.85 S2* − 16.40 S3* − 21.48 S4* + 2.62 Freq:S1 + 1.49 Freq:S2 + 1.31 Freq:S3 + 1.50 Freq:S4 − 0.95 Freq$^2$:S1 − 0.07 Freq$^2$:S2 − 0.07 Freq$^2$:S3 − 0.07 Freq$^2$:S4 |
| **Oxic mineralization** | y = Intercept + a · Freq + b · Freq$^2$ + c · Stat + d · Freq:Sed + e · Freq$^2$:Sed |
| | y = 23.29 - 0.12 Freq − 0.20 Freq$^2$ - 0.21 S1 − 16.74 S2 − 11.28 S3 − 17.17 S4 + 7.63 Freq:S1 + 0.58 Freq:S2 − 0.38 Freq:S3 + 0.91 Freq:S4 − 0.90 Freq$^2$:S1 + 0.06 Freq$^2$:S2 + 0.19 Freq$^2$:S3 + 0.01 Freq$^2$:S4 |
| **Anoxic mineralization** | y = Intercept + a · Freq + b · Freq$^2$ + c · Stat + d · Freq:Sed + e · Freq$^2$:Sed |
| | y = 6.57 − 1.23 Freq + 0.14 Freq$^2$ + 51.14 S1 − 5.53 S2 − 5.67 S3 − 5.03 S4 − 8.27 Freq:S1 + 0.80 Freq:S2 + 1.12 Freq:S3 + 0.59 Freq:S4 + 0.55 Freq$^2$:S1 − 0.09 Freq$^2$:S3 − 0.13 Freq$^2$:S3 - 0.07 Freq$^2$:S4 |
| **Denitrification** | y = Intercept + a · Freq + b · Freq$^2$ + c · Gear + d · Stat + e · Freq:Sed + f · Freq$^2$:Sed |
| | y = 0.09 − 0.0034 Freq - 0.00035 Freq$^2$ − 0.00038 TicklerGear + 4.14 S1 + 0.44 S2 + 0.55 S3 + 0.73 S4 + 1.26 Freq:S1 - 0.11 Freq:S2 + 0.06 Freq:S3 − 0.03 Freq:S4 − 0.18 Freq$^2$:S1 + 0.01 Freq$^2$:S2 − 0.02 Freq$^2$:S3 - 0.01 Freq$^2$:S4 |
| | *S1 = MudH, **S2 = FineL, ***S3 = Coarse, ****S4 = MudL |



**Table 7: Overview of effect of increasing trawling intensities for tickler gears, and pulse gears (columns) on the total organic matter mineralization, and the different mineralization processes. The baseline scenario (frequency = 0) is displayed as the absolute rate (mmolC m$^{-2}$ d$^{-1}$), increasing frequencies (1 – 5 y$^{-1}$) are shown as % change of the baseline rate. Mineralization rates are the average annual concentrations.**

| | | Tickler | | | | | | Pulse | | | | | |
| | | Rate | ------- % Change ------- | | | | | Rate | ------- % Change ------- | | | | |
| | station | 0 | 1 | 2 | 3 | 4 | 5 | 0 | 1 | 2 | 3 | 4 | 5 |
| **Total** mmol C m$^{-2}$ d$^{-1}$ | Coarse | 13.6 | -5 ± 0 | -8 ± 13 | -12 ± 10 | -19 ± 9 | -22 ± 10 | 13.6 | -5 ± 0 | -7 ± 7 | -13 ± 11 | -17 ± 7 | -21 ± 5 |
| | FineL | 8.1 | -5 ± 3 | -13 ± 15 | -20 ± 12 | -21 ± 5 | -27 ± 5 | 8.1 | -6 ± 4 | -8 ± 8 | -18 ± 7 | -22 ± 5 | -24 ± 4 |
| | FineH | 30.0 | -7 ± 2 | -15 ± 16 | -17 ± 13 | -24 ± 8 | -27 ± 7 | 30.0 | -7 ± 3 | -11 ± 13 | -17 ± 8 | -23 ± 8 | -29 ± 6 |
| | MudL | 8.6 | -5 ± 2 | -15 ± 19 | -18 ± 10 | -23 ± 8 | -27 ± 6 | 8.6 | -6 ± 4 | -11 ± 13 | -17 ± 9 | -21 ± 5 | -25 ± 7 |
| | MudH | 85.0 | -7 ± 2 | -12 ± 12 | -20 ± 10 | -22 ± 5 | -26 ± 6 | 85.0 | -6 ± 3 | -11 ± 9 | -17 ± 8 | -25 ± 8 | -26 ± 5 |
| **OxicMin** mmol C m$^{-2}$ d$^{-1}$ | Coarse | 12.0 | -4 ± 1 | -8 ± 13 | -12 ± 10 | -18 ± 9 | -21 ± 10 | 12.0 | -4 ± 1 | -7 ± 7 | -13 ± 10 | -17 ± 7 | -21 ± 5 |
| | FineL | 6.5 | 8 ± 4 | -1 ± 17 | -8 ± 14 | -7 ± 6 | -15 ± 6 | 6.5 | 7 ± 4 | 5 ± 9 | -5 ± 8 | -10 ± 6 | -12 ± 5 |
| | FineH | 23.3 | -1 ± 2 | -10 ± 17 | -13 ± 14 | -20 ± 8 | -23 ± 7 | 23.3 | -2 ± 3 | -6 ± 14 | -13 ± 9 | -19 ± 9 | -25 ± 6 |
| | MudL | 6.1 | 11 ± 3 | 0 ± 22 | -2 ± 12 | -7 ± 10 | -11 ± 7 | 6.1 | 10 ± 4 | 5 ± 15 | -1 ± 10 | -5 ± 6 | -9 ± 8 |
| | MudH | 23.1 | 40 ± 10 | 50 ± 19 | 48 ± 21 | 54 ± 15 | 56 ± 12 | 23.1 | 32 ± 5 | 46 ± 12 | 45 ± 16 | 48 ± 22 | 56 ± 15 |
| **Anoxic Min** mmol C m$^{-2}$ d$^{-1}$ | Coarse | 0.9 | -20 ± 14 | -24 ± 17 | -20 ± 16 | -23 ± 15 | -24 ± 12 | 0.9 | -23 ± 12 | -20 ± 14 | -18 ± 19 | -25 ± 10 | -24 ± 8 |
| | FineL | 1.0 | -77 ± 3 | -79 ± 4 | -79 ± 2 | -78 ± 2 | -80 ± 2 | 1.0 | -75 ± 5 | -77 ± 3 | -79 ± 2 | -80 ± 2 | -80 ± 1 |
| | FineH | 6.6 | -26 ± 4 | -31 ± 13 | -33 ± 11 | -38 ± 6 | -41 ± 6 | 6.6 | -26 ± 5 | -27 ± 11 | -32 ± 8 | -38 ± 8 | -42 ± 5 |
| | MudL | 1.5 | -74 ± 1 | -78 ± 5 | -80 ± 3 | -82 ± 3 | -83 ± 2 | 1.5 | -73 ± 2 | -76 ± 4 | -80 ± 3 | -81 ± 2 | -83 ± 2 |
| | MudH | 57.7 | -27 ± 6 | -39 ± 9 | -49 ± 10 | -54 ± 7 | -61 ± 7 | 57.7 | -22 ± 3 | -36 ± 9 | -43 ± 8 | -57 ± 7 | -60 ± 5 |
| **Denitrification** mmol C m$^{-2}$ d$^{-1}$ | Coarse | 0.6 | 11 ± 14 | 9 ± 20 | -2 ± 20 | -14 ± 14 | -20 ± 14 | 0.6 | 11 ± 16 | 8 ± 19 | -6 ± 14 | -8 ± 17 | -19 ± 9 |
| | FineL | 0.5 | -25 ± 7 | -42 ± 15 | -57 ± 13 | -70 ± 9 | -74 ± 8 | 0.5 | -19 ± 6 | -30 ± 11 | -49 ± 9 | -60 ± 11 | -68 ± 9 |
| | FineH | 0.1 | 0 ± 22 | -11 ± 22 | -11 ± 22 | -22 ± 11 | -22 ± 11 | 0.1 | 0 ± 22 | -11 ± 22 | -11 ± 22 | -22 ± 11 | -33 ± 11 |
| | MudL | 0.8 | 1 ± 4 | -13 ± 20 | -22 ± 12 | -31 ± 9 | -35 ± 6 | 0.8 | 4 ± 5 | -6 ± 15 | -20 ± 12 | -28 ± 9 | -33 ± 6 |
| | MudH | 4.2 | 35 ± 15 | 46 ± 26 | 39 ± 20 | 48 ± 17 | 50 ± 9 | 4.2 | 26 ± 15 | 42 ± 17 | 41 ± 17 | 42 ± 19 | 49 ± 11 |
