# Peer review of "Impact of bottom trawling on sediment biogeochemistry: a modelling approach"

_Biogeosciences, 2020_

## Referee Comment (RC1) · Sarah Paradis (Referee) · 27 Oct 2020

General comments:

De Borger et al. address in this manuscript the potential biogeochemical effects of different bottom trawling gear on different seafloor environments. They assess the effect of bottom trawling in different mineralization pathways through a modelling approach. The paper presents several editing mistakes (reference errors, table not properly ordered), and certain aspects in the methodology should be explained in more detail. Nevertheless, the authors provide novel and sound results of the biogeochemical effects of bottom trawling on the seafloor, revealing that different trawling gear types with different penetration depths have similar effects on sediment biogeochemistry. The

authors highlight that the greatest impact on the seafloor occurs just by trawling once, and that management strategies should be aimed at limiting spatial extension of bottom trawling grounds rather than improving technical configurations on trawling gears.

The following remarks and suggestions should help clarify certain aspects of the manuscript.

Specific comments:

1. How would the results change with bottom trawling gear with even greater penetration on the seabed? For instance, bottom trawl doors can penetrate tens of centimeters deep in the sediment, an order of magnitude greater than those you report in this study for the tickler chain (3.2 cm) and pulse wing (1.6 cm) gear types.

2. Section 2.1.2 Model parametrization explains the different sediment cores collected to conduct the modelling approach. In this section, the authors redirect the reader to unpublished papers (De Borger et al., 2020; Toussaint et al., 2020) for additional information of the sampling procedure. Please refrain from referring to unpublished papers and provide all the necessary information of the sampling technique (e.g. subsampling boxcores, sediment analyses, frequency of analyses?). Similarly, please provide any necessary explanations of the steady state fitting procedure (lines 112-113) that is described in the manuscript that is in-preparation (De Borger et al. 2020). Finally, please correct the sources described in Table 2, and explain how the sinusoidal flux of C illustrated in Figure 3A was obtained.

3. Mortality of benthic fauna affects the biogeochemical modelling approach by reducing bioturbation rates. Please comment and address other aspects of how this higher mortality can affect sediment biogeochemistry. For instance, with fewer benthic fauna, wouldn't that increase the burial of organic matter on the seafloor? Moreover, would the electrodes in pulse gear cause greater mortality of benthic species?

4. You mention how the low permeability of muddy sediments affect mineralization

pathways (line 308). Could you elaborate here the effect on coarser and more permeable sediment?

5. Based on your results, how would bottom trawling affect the composition of OC (labile, semi-labile content of OC?). For instance, in line 210-211, do you mean that trawling increases the relative amount of labile OC?

6. Could you elaborate on how your results differ from those modelled by Duplisea et al. (2001)?

7. Finally, this manuscript would benefit by addressing, based on your results, the effects of bottom trawling in different margins. Moreover, considering that bottom trawling now generally occurs at ∼500 m water depth (Watson and Morato, 2013, Fisheries Research), what would be the biogeochemical impacts of bottom trawling?

Technical corrections:

Line 81-82. Is the incoming flux in all model scenario the same, irrespective of the OM content of the seafloor?

Line 120. Note that you refer to Figure 3 before referring to Figure 2. Please amend this.

Line 135, 137. Provide the definition of mud.

Line 114, 181. Please refer to the tables in order of appearance. Currently in the text, you refer to the tables in the following order of appearance: Table 1, 2, 4, 5, 3. Please change the order of the tables so that they are referred to sequentially.

Line 157. Pitcher et al. (2020) do not talk about the penetration depths of different trawling gear. Where do you get these values?

Line 160; 217. Please correct the missing reference source here.

Line 181-183. Specify that the higher remineralization rates here are due to the higher

organic matter content of these sites (MudH, FineH).

Line 195-196. This sentence refers to both Figure 4 and Figure 5.

Lines 231-240. Refer to Table 7 in the text.

Line 238-239. "Trawling frequencies of $1 – 2$ y-1 caused increases in the denitrification for the Coarse sediment, and FineH and MudL". Considering the error, denitrification in these sediment types did not increase in these trawling frequencies, only for the MudH sediment, as you mention later on.

Lines 241-245. Where is the data that illustrates this?

Lines 334-335. Just earlier, you mention increases in denitrification, but here you mention that your results are comparable to those of Ferguson et al. (2020) who observed decreases in denitrification. Please clarify that your results, with the exception of MudH, also present decreases in denitrification.

Line 361. Change "resuspended silt" to "resuspended fine-grained sediment", since both silt and clay can be resuspended and preferentially advected away from bottom trawling grounds (Martin et al., 2014, PLoS ONE). You focus on how this kind of effect varies with different bottom trawling gear types, but could you elaborate on how this would affect sediment biogeochemistry?

Line 367-368. Modify "deep and shallow penetrating gear types", since 3 cm of penetration is not considered to be deep.

Table 2: Please provide definition of the acronyms (MGS, SD.9, SD.1, VFines). Could you also give the proportion of sand (2 mm – 63 $\mu$m) content (%)? Provide also the grain size ranges to define the proportion of silt and fine-grained sediment. If fine-grained sediment is consistent with the grain size range of clays (< 4 $\mu$m), please consider giving it this terminology.

Figure 3: A. Consider using "Day of year" in the axis title. If this carbon flux consists of

only DIC, why is it applicable to both organic and inorganic fluxes? In lines 119-120 you refer to this figure as the carbon flux. Please provide more information. B. Where did the data from this figure come from? C. According to Eq. 3, biota depletion depends on both penetration depth and sediment grain size, but only penetration depth (which varies with different gear) is shown in this figure. Please include the effect of sediment grain size in this figure, since it is a very important parameter, rather than giving it in Table 3.

Consider converting Tables 5 and 7 as figures, since the changes in terms of frequency, gear type, and sediment type would be more visual.

Figure S1: The fitted red lines for FineL-oxygen, MudH-nitrate are not adjusted to the concentration profiles measured.

---

## Referee Comment (RC2) · Antonio Pusceddu (Referee) · 12 Nov 2020

Antonio Pusceddu (Referee)

apusceddu@unica.it

General comments The manuscript by De Borger et alii addresses the still debated (partially controversial) issue of bottom trawling impacts on sediment biogeochemistry. Using a modelling approach based on the implementation of an early digenesis model (OMEXDIA), parametrized for five different settings in the North Sea, this study provides evidence that increasing bottom trawling frequency can result in increased oxygen and nitrate concentrations in surface sediments, counterbalanced by a net loss of ammonium and organic C down to 10 cm depth in the sediment. The model output showed also a net decrease (up to 28%) of total mineralization rates because of bottom trawling, even in the case of trawling gears penetrating just the top few centimetres of the sedimentary column. Sediment resuspension and C removal associated with bottom trawling were identified as the most probable putative causes for the decrease of mineralization rates in trawled sediments.

The authors have been successful in providing a clear background of the available knowledge on the topic and gave an honest and complete credit to previous studies dealing with the same issue. This allowed them to provide a clear identification of the aim of their study, starting from the to-the-point and concise title. Also, the abstract is informative and mirrors accurately the contents of the manuscript and its main conclusions. The structure of the manuscript is clear and the language fluent as well (at least for a not-mother language speaking reader like me). The initial assumptions are valid, the modelling methods and tools used for data analysis are also appropriate and properly used. All figures are clear and necessary and the tables are all informative ((but see below the technical suggestions).

Overall, this elegant study represents an important contribution to the specific topic, which definitely fits the scope of BG. The paper provides robust cues which, altogether, allow to (partially) unravel the contradictory results obtained by different studies addressed to identify the effects of bottom trawling on sediment biogeochemistry.

Specific comments Overall, the results support the interpretation of the differences in biogeochemical responses to bottom trawling carried out with different gears, in different environmental contexts and with variable frequency.

Despite this, I guess that the authors should make an effort to better explain how the bioturbation rates were calculated. As far as I can see, this information is not fully included in the manuscript and this could be crucial to understand how much and whether the abundance, biomass and (functional) diversity of the benthos in the parameterisation sites have been taken into account. In my opinion, this information is also necessary because the results, though contextualized according to the different sediment granulometry of the experimental sites and C fluxes, do not seem to have been analysed eliminating the covariate effect of water depth, which controls benthos

abundance, biomass and diversity, which, in turn, could respond differentially to bottom trawling disturbance (as indeed postulated in the discussion).

As mineralization rates, as correctly postulated in the manuscript, are dependent also on the relative importance of refractory and labile/semi-labile fractions of OC, it could be interesting to see addressed the effects of bottom trawling on the two fractions, though this could be the object of a "sister" manuscript.

Technical suggestions: Despite all of the figures are necessary and informative, a (qualitative) graphic panel of differences and tendencies of mineralization rates along the trawling frequency gradient in the different environmental contexts (sediment type and depth) would help a lot the general reader to recap the results.

The number of tables could be reduced, moving some of them in the supplement material.

Figures' and Tables' numbering (and their order of reference in the main text) need an accurate check and correction.

---

## Author Comment (AC1) · 9 Dec 2020

We thank the reviewer for the time taken to read and comment on the manuscript. Most of the suggestions and requests for additional clarification were implemented in the manuscript, including the addition of a new figure (which was also suggested by the other reviewer). We did choose to keep the reference to unpublished works, from which data was used to support the modelling setup (Toussaint et al., in review, De Borger et al., in review). Both works are in the reviewing process (one requiring major revisions, mostly related to framing the novelty of the research, the other resubmitted after minor revisions). However, we understand the need for clarity in this manuscript itself, and opted to add additional information about the sampling procedure in the supplementary data, and to extend the explanation of the fitting procedure. In the

attachment, each remark is answered individually. With these changes, we hope to have replied to this review in a satisfactory manner.

Please also note the supplement to this comment:
https://bg.copernicus.org/preprints/bg-2020-328/bg-2020-328-AC1-supplement.pdf

---

## Author Response (AR1)

**Reply to the reviewers**

We thank the reviewers for the time taken to read and comment on the manuscript. Most of the suggestions and requests for additional clarification were implemented in the manuscript, including the addition of a new figure (which was also suggested by the other reviewer).

We did choose to keep the references to unpublished works, from which data was used to support the modelling setup (Toussaint et al., *in review*, De Borger et al., *in press*). In the meantime, Toussaint et al. has been resubmitted after major revisions (mostly related to the framing of the research), and De Borger et al. has been accepted for publication. However, we understand the need for clarity in this manuscript itself, and opted to add additional information about the sampling procedure in the supplementary data, and to extend the explanation of the fitting procedure. We were happy to elaborate on the methodology of deriving bioturbation rates, and implement the suggested changes to figures and tables. In addition, we took up the suggestion to make a sort of summary figure, to serve as the figure for the abstract.

The changes were applied in track changes in the manuscript, and copied in this reply to individual answers. With these changes, we hope to have replied to this review in a satisfactory manner.

**REVIEWER #1**

1. **Reviewer's comment:** How would the results change with bottom trawling gear with even greater penetration on the seabed? For instance, bottom trawl doors can penetrate tens of centimeters deep in the sediment, an order of magnitude greater than those you report in this study for the tickler chain (3.2 cm) and pulse wing (1.6 cm) gear types.

   **Reply:** Assuming that the eroded layer and the mixing depth scale with increasing total penetration depth of these other gears, the depletion effects should become even more pronounced. In a single trawling event more of the (reactive carbon rich) top sediment layer would be removed, there would be a higher mortality of organisms, and more of the nutrient build-up would be removed. For the trawling gears we investigated the depth distributions happened to be more or less known (please see our reply to comment #12 + the works of Depestele et al., 2016, 2019 on the eroded layer dynamics), which made it possible to realistically parametrize the modelling mechanics. To implement this remark (more concisely), as well as remark #23, we chose to restructure section 4.3 in the discussion entirely.

   Please note that we also added a reference to the differences in catch-efficiency between pulse gears and tickler-chain gears, so that we do not seem to suggest anymore that improvements to fishing gear do not matter.

   This section now reads:

   " *In our work the differences with respect to organic matter mineralization dynamics between gear types with differing penetration depths (e.g. 3.2 ± 1.2 cm vs. 1.6 ± 0.6 cm in mud) mostly remained suggestive, rather than statistically conclusive. The fishing gear type was only included as a significant predictor for the denitrification rates, with a small coefficient (-0.00038; Table 5). This is because the freshly deposited stocks of organic carbon are present near the sediment surface, and any gear that penetrates the sediment impacts this layer, especially for multiple trawling events per year (Figure 5, Table 5). This indicates that only a thin layer of surface sediment needs to be disturbed to generate significant biogeochemical changes (Dounas et al., 2005). Many biogeochemical processes are mediated by the dynamics of oxygen near the sediment-water interface, which itself is influenced by the composition, and permeability of the sediment. Permeable sediments allow for a deeper flow of oxidized bottom water into the sediment matrix, increasing the available oxygen for oxic mineralization processes (Huettel and Gust, 1992). Cohesive sediments on the other hand, mostly lack such advective transport (Huettel and Gust, 1992). Instead the exchange of solutes between sediment and the overlying water is diffusion dominated (molecular and biogenic), resulting in a less deep oxic zone at comparable organic matter loads, and an increased importance of other oxidants (e.g. $MnO_2$, $FeOOH$, $SO_4^{2-}$). A shift towards fining (an increased proportion of finer grain size classes) has been described in certain trawled areas, with expected consequences for sediment biogeochemistry, such as an increased rate of sulphate reduction (Trimmer et al., 2005). But the opposite occurs just as well (Depestele et al., 2019; Mengual et al., 2019; Tiano et al., 2019). In these cases resuspended fine-grained material is exported away from the trawling site, leaving a coarsened trawling track, with the results subtly different between gear types (Depestele et al., 2019; Tiano et al., 2019). This means that the effects of fishing gears on grain size sorting should be better characterized for various sediment types, to constrain the uncertainty around predictions of gear impacts on sediment functioning.*
   *This does not imply that the penetration depth is irrelevant. Other studies have reported clear positive effects of reducing the penetrations depths of fishing gears, such as decreased sediment mobilization and*

*homogenization, and reduced organic matter depletion (Depestele et al., 2019; Tiano et al., 2019). Conversely, assuming that the eroded layer and the mixing depth scale with more deeply penetrating gears than those tested here, the depletion effects should become more pronounced. In a single trawling event more of the (reactive carbon rich) top sediment layer would be removed, there would be a higher mortality of* 65 *organisms, and more of the nutrient build-up would be removed.*

*Aside from the penetration depth, largest impacts occurred when increasing the trawling frequency from 0 to 1 trawling event per year, and the response of mineralization processes to increased trawling frequency was often non-linear, making them more difficult to predict. This would imply that management strategies aimed at maintaining the ecosystem functions provided by shelf sediments should be focused on spatial controls,* 70 *bottom impact quotas, and effort control of trawling gears that per definition require contact with the bottom to catch commercially viable target species (McConnaughey et al., 2020). An effective strategy limits the impacted surface area, and allows carbon stocks and faunal communities in the sediment to recover from a disturbance, resulting in the recovery of vital biogeochemical functions such as denitrification and carbon burial. This includes technical adaptations to improve catch-efficiency. Whereas our study only focused on* 75 *direct head-to-head comparisons between the two gear types, pulse trawls are associated with lower spatial footprints due to their relatively higher catch efficiencies compared to beam trawls (Poos et al., 2020; Rijnsdorp et al., 2020; ICES, 2020).*

*Shifting the fishing effort from peripheral areas to core fishing grounds would also reduce the area where the top sediment layer is removed on a regular basis, along with the associated reduction in mineralization of* 80 *organic matter. This can be achieved through time-restricted bottom trawling, in which fishing grounds are closed off temporarily. Similarly, areas with high denitrification rates, crucial for eutrophication mitigation can be closed to trawling completely, as done in other regions of the world (Ferguson et al., 2020). Site-specific conditions such as rates of biogeochemical recovery and sedimentation rates need to be known to determine the resilience of ecosystems to trawling, and fine-tune management plans (Paradis et al., 2019)."*

85

**2. Reviewer's comment:** Section 2.1.2 Model parametrization explains the different sediment cores collected to conduct the modelling approach. In this section, the authors redirect the reader to unpublished papers (De Borger et al., 2020; Toussaint et al., 2020) for additional information of the sampling procedure. Please refrain from referring to unpublished papers and provide all the necessary information of the sampling technique (e.g. 90 subsampling boxcores, sediment analyses, frequency of analyses?). Similarly, please provide any necessary explanations of the steady state fitting procedure (lines 112-113) that is described in the manuscript that is in-preparation (De Borger et al. 2020). Finally, please correct the sources described in Table 2, and explain how the sinusoidal flux of C illustrated in Figure 3A was obtained.

95 **Reply:** With both manuscripts that are referred to still in review, we agree that additional information on the sampling procedure benefits the manuscript. Since in time these articles will be published as well, we therefor opted to add this rather lengthy detailing of the sampling and processing procedure in the supplementary information, and not in the main body of the text where we now refer to both manuscripts in review, as well as the supplementary information.

100

The text added to the supplement is shown at the bottom of this reply. More information on the steady-state fitting procedure was added to the main body of text as requested (by both reviewers) in lines 116-134: "*Model parameters included both measured concentrations in the bottom water, as well as process rate parameters that were derived following a 2-step steady state fitting procedure (Table 3). Using the measured DIC flux as* 105 *the upper boundary carbon input flux, the $O_2$ flux and porewater profiles of $O_2$, $NO_3^-$, and $NH_4^+$ were first fitted manually by tweaking a limited set of model parameters. The degradation rate of refractory material (rSlow), and the biodiffusivity constant Db were constrained by fitting $NH_4^+$, and $O_2$ profiles. Mechanistically, decreasing the bioturbation rate reduces the build-up of $NH_4^+$ with depth, increases the oxygen penetration depth, and changes the shape of the $NO_3^-$ profile (deepening the $NO_3^-$ peak). Whereas the degradation rate of* 110 *the refractory organic matter also impacts deep $NH_4^+$ concentrations, it has a larger effect on the shape of the $NH_4^+$ profile, with lower degradation rates causing a more gradual build-up with sediment depth. Subsequently, parameters affecting the $NO_3^-$ and $NH_4^+$ profiles were tuned (the nitrification rate rnit, and denitrification constants ksNO3denit and kinO2denit). Higher nitrification rates increase the build-up of $NH_4^+$, and increase concentrations of $NO_3^-$, typically producing a nitrate concentration peak within the oxic* 115 *zone. The shape of the oxygen profiles further constrained the oxidation rate of oxygen demanding units (ODU's), and inhibition constants for anoxic mineralization (kinO2anox, kinNO3anox).*

*The manual fitting was followed by a constrained parameter fitting using an optimization algorithm. In this second step, the fitted parameters were allowed to vary in a range ± 10 % around the manually fitted parameter values. Also the DIC fluxes were refitted within a narrow range (0.98 -1.02 of measured value),* 120 *to allow freedom to the fitting algorithm. A random-based minimization algorithm (Price, 1977) implemented*

*in the R package FME (Soetaert and Petzoldt, 2010) was used. This algorithm pseudo-randomly sampled the parameter space, until the parameter set was found which returned the minimal model cost, the latter defined as the sum of variable costs (modeled - measured values), scaled using the mean - standard deviation relation determined for each nutrient.*"

125

The incoming carbon flux differs for each location. This is now clarified in section 2.1.2., lines 119 – 121. "*A sinusoidally varying carbon deposition flux with the model derived carbon flux (Cflux, Table 3) as the annual average, and imposing an amplitude of 1 was used as the upper boundary organic carbon flux (Figure 2 A). This resulted in differing organic carbon fluxes for each location.*" The second sentence was added, as well

130    as the word "organic" in the first to aid with clarifying remark #23. Note also that Figure 3 and Figure 2 were switched to comply with remark #9.

**3.    Reviewer's comment:** Mortality of benthic fauna affects the biogeochemical modelling approach by reducing bioturbation rates. Please comment and address other aspects of how this higher mortality can affect sediment

135    biogeochemistry. For instance, with fewer benthic fauna, wouldn't that increase the burial of organic matter on the seafloor? Moreover, would the electrodes in pulse gear cause greater mortality of benthic species?

**Reply:** With fewer bioturbating fauna, organic carbon will remain near the sediment surface for longer, exposed to oxygen. Organic matter will thus be degraded surficially through oxic mineralization processes. It

140    is partly the burial of organic matter by bioturbators that exposes relatively fresh organic matter to sub- anoxic sediment conditions in deeper sediment horizons. After a certain arbitrary depth (e.g. 1 cm), you can then speak of burial.
Existing literature does not indicate an increased mortality of benthic fauna due to electrodes (van Marlen et al., 2009; Soetaert et al., 2015, 2016; Murray et al., 2016; ICES, 2020). The studies all show no, or very limited

145    benthos mortality through electric exposure and only short term behavioural effects on macrobenthic species such as *Ensis sp.*, *Pagurus bernhardus*, *Asterias rubens*, *Spisula sp.*, *Nereis sp*. From our own direct observations in laboratory experiments (unpublished), we know that *Arenicola marina*, first of all survive electric pulses, and secondly recover their normal behaviour within 10 – 15 minutes after exposure.
As this is a good point that other readers of this manuscript may have, we have added the above references to

150    the materials and methods section (line 144-) to justify why the depletion mechanics were the same for both gears. "*Electrical pulses were assumed to not affect benthos mortality in addition to the physical effects, since current available research shows very limited to no increased mortality by electrical pulses when compared to control situations (van Marlen et al., 2009; Soetaert et al., 2015, 2016; Murray et al., 2016; ICES, 2020).*"

155    **4.    Reviewer's comment:** You mention how the low permeability of muddy sediments affect mineralization pathways (line 308). Could you elaborate here the effect on coarser and more permeable sediment?

**Reply:** We chose to implement this comment in lines 335-338, where a short discussion of the Coarse sediment was already present. These lines now read: "*Larger pore-spaces in these sediments allow for bottom*

160    *water to penetrate more deeply into the sediment matrix, bringing oxygen and other reactants into deeper sediment layers. As a result, oxygen often fuels rapid mineralization in coarse grained sediments* (Huettel and Rusch, 2000; Ehrenhauss et al., 2004). *The increasing trawling frequency in coarse sediments thus had little effect on oxygen penetration, and nitrate concentrations only marginally increased (Figure 7: A-B), resulting in minor changes to mineralization pathways such as denitrification on average, although instantaneous*

165    *effects could be prominent (Figure 7 C)*".

**5.    Reviewer's comment:** Based on your results, how would bottom trawling affect the composition of OC (labile, semi-labile content of OC?). For instance, in line 210-211, do you mean that trawling increases the relative amount of labile OC?

170

**Reply:** That is indeed the correct interpretation of our result, valid for chronic bottom trawling. In our model, bottom trawling causes a depletion of the carbon build-up in the sediment. As organic carbon generally becomes less reactive with age (~ depth in the sediment), the total organic carbon pool becomes more reactive when less organic carbon gradually builds up to deeper layers. In the most extreme case (as an example), there

175    is almost no organic carbon in the sediment due to continuous, chronic resuspension. If fresh organic carbon is then deposited on the sediment-water interface, the labile organic carbon will proportionally make up a considerable part of the total organic carbon pool.
Since also the other reviewer noted interest in changes to the reactive carbon pool, we added the figure (please see below) to the supplementary information, and refer to this figure on line 235: "*In FineL, MudL and MudH*

*the ratio of labile organic carbon (FDET) to semi-labile organic carbon (SDET) increased between 25 and 34 % (Figure S3, supplement).*"

[Figure]

**Figure S 1: Annually averaged modelled quality of the reactive organic carbon pool in the surface sediments (note different depths on y-axis between figures for visualization purposes). The carbon quality (x-axis) is represented as the proportion of fast degrading detritus (FDET, labile org. C) in the summed labile and semi-labile org. C pool (FDET + SDET). Black dotted line is the 0 trawl default, full and dotted coloured lines are tickler and pulse gear respectively, with increasing trawling frequencies as different colours.**

6.  **Reviewer's comment:** Could you elaborate on how your results differ from those modelled by Duplisea et al. (2001)?

**Reply:** The main differences between the work of Duplisea et al., 2001 and our work stem from the different models that were used. We used diagenetic reactive transport model in which sediment depth is the spatial dimension, and the biogeochemical reactions that connect organic carbon and solutes are explicitly represented (please see materials and methods). Duplisea et al., used a food-web model in which different components (e.g. organic carbon pool, macrofauna, bacteria, meiofauna) exchange carbon with one-another. Additionally, this model contains no spatial dimension. So in fact, these are two completely different models, from which output is very different and difficult to compare.

That being said, it is interesting that two findings (the stabilizing effect of macrofauna in terms of carbon, and an increased contribution of oxic/aerobic mineralization) are highlighted in both works, though also with different underlying mechanistic explanations.

Besides this, there is not much to compare, as we reported on biogeochemical variables not included in the work of Duplisea et al., such as porewater profiles, relative changes of additional mineralization processes, the effects of different settings, the erosion-mixing effect,…

We have added a short note on the comparison with the work of Duplisea et al. (2001) on lines 301- : "*The comparison with the work of Duplisea et al. (2001) is in fact remarkable, as Duplisea et al. (2001) used a food-web based model to investigate changes to carbon cycling, whereas a diagenetic model was used in this work. Both very different approaches, that both highlight a shift to more oxic mineralization, and the importance of benthic fauna as a stabilizing factor.*"

7.  **Reviewer's comment:** Finally, this manuscript would benefit by addressing, based on your results, the effects of bottom trawling in different margins. Moreover, considering that bottom trawling now generally occurs at _500 m water depth (Watson and Morato, 2013, Fisheries Research), what would be the biogeochemical impacts of bottom trawling?

**Reply:** For sure, we expect the effects in deeper waters (continental slope – deep sea) to be even greater, given the higher vulnerability of species (long-lived, slower growth), more cohesive seidments, as well as decreased deposition rates of organic matter. As an extreme example, recent studies on the effects of deep sea mining show that effects of a mining trawl (10 – 20 cm of mixing) to sediment morphology (Gausepohl et al., 2020), as well as food-web structure remain visible after 26 (!) years (de Jonge et al., 2020), with also microbially regulated biogeochemical functions still impaired (Vonnahme et al., 2020).

We have rewritten lines 306-314 to accommodate this information concisely. "*Also, OM depletion as a result of long term fishing has been reported, even at water depths beyond 500 m (Martín et al., 2014; Pusceddu et al., 2014; Paradis et al., 2019), where comparisons between trawled and untrawled sites yielded a difference in OC between 20 and 60 % (Paradis et al., 2019), or 60 – 100 % of the daily input flux of organic carbon was removed from sediments by trawling (Pusceddu et al., 2014). In fact these deep-water sediments are particularly sensitive to trawling disturbances, a concerning fact given the steady expansion of fishing practices into deeper waters in recent decades (Morato et al., 2006; Puig et al., 2012; Watson and Morato, 2013). Deep-water species communities are slow-growing and thus recover slowly, organic matter deposition rates are low, and finer-grained sediments of the deep are easily resuspended following a trawl passage (Norse et al., 2012; Mengual et al., 2016). All three of these factors increase the impacts of trawling events on organic matter cycling in the model presented in this work.*"

8. **Reviewer's comment:** Line 80-81. Is the incoming flux in all model scenario the same, irrespective of the OM content of the seafloor?

**Reply:** The incoming carbon flux differs for each location. This is now clarified in section 2.1.2., lines 119 – 121. "A sinusoidally varying carbon deposition flux with the model derived carbon flux (*Cflux*, Table 3) as the annual average, and imposing an amplitude of 1 was used as the upper boundary organic carbon flux (Figure 2 A). This resulted in differing organic carbon fluxes for each location." The second sentence was added, as well as the word "organic" in the first to aid with clarifying remark #23.

9. **Reviewer's comment:** Note that you refer to Figure 3 before referring to Figure 2. Please amend this.

**Reply:** Thank you for noticing this. This has now been rectified, by switching the positions (and numbering) of figures 2 and 3.

10. **Reviewer's comment:** Line 135, 137. Provide the definition of mud.

**Reply:** The definition of mud has been added between brackets on line 137, which now reads: "*It was calculated based on the total gear penetration depth (TPD, i.e. the sum of the eroded layer depth and the penetration depth, in cm), and the mud content (% mud¸ particles < 63 µm) of the sediment, as described in Eq. 3Error! Reference source not found. derived from Sciberras et al. (2018).*"

11. **Reviewer's comment:** Please refer to the tables in order of appearance. Currently in the text, you refer to the tables in the following order of appearance: Table 1, 2, 4, 5, 3. Please Change the order of the tables so that they are referred to sequentially

**Reply:** This has now been rectified, by changing the order of the tables, and moving tables 5 and 7 to supplementary information.

12. **Reviewer's comment:** Line 157. Pitcher et al. (2020) do not talk about the penetration depths of different trawling gear. Where do you get these values?

**Reply:** A manuscript by Pitcher et al., co-authored by an author of this article (Rijnsdorp, A.) is currently in review. In this manuscript, trawling impacts on seabed habitat status of different regions is assessed. As part of this work, current knowledge on average penetration depths of trawling gears were aggregated, and this information was shared with us to prepare our own work. So, we have changed the reference to (pers. comm.).

13. **Reviewer's comment:** Line 160; 217. Please correct the missing reference source here.

**Reply:** Thank you for noticing this, these referencing errors have been corrected.

**14. Reviewer's comment:** Line 181-183. Specify that the higher remineralization rates here are due to the higher organic matter content of these sites (MudH, FineH).

**Reply:** A reference to the higher organic matter deposition fluxes between the low – high nutrient sites has been added to this section: *"The two muddy stations had either very high or very low total mineralization rates (MudH: 82 mmol C $m^{-2}$ $d^{-1}$, MudL: 8.5 mmol C $m^{-2}$ $d^{-1}$), and similar for the two fine sandy stations (FineH: 30 mmol C $m^{-2}$ $d^{-1}$, FineL: 8.1 mmol C $m^{-2}$ $d^{-1}$), related to the difference in organic matter deposition between nearshore – offshore locations (Table 3)."*

**15. Reviewer's comment:** Line 195-196. This sentence refers to both Figure 4 and Figure 5.

**Reply:** The additional reference has been added.

**16. Reviewer's comment:** Lines 231-240. Refer to Table 7 in the text.

**Reply:** References to the corresponding figure and table have been added,

**17. Reviewer's comment:** Line 238-239. "Trawling frequencies of $1-2$ $y^{-1}$ caused increases in the denitrification for the Coarse sediment, and FineH and MudL". Considering the error, denitrification in these sediment types did not increase in these trawling frequencies, only for the MudH sediment, as you mention later on.

**Reply:** This is indeed correct. We have rectified line 238-239 as follows: *"Trawling frequencies of $1-2$ $y^{-1}$ did not consistently alter denitrification for the Coarse sediment, and FineH and MudL."*

**18. Reviewer's comment:** Lines 241-245. Where is the data that illustrates this?

**Reply:** This information is derived from the denitrification rates now shown in the supplement, but are not represented separately in a table.

**19. Reviewer's comment:** Lines 334-335. Just earlier, you mention increases in denitrification, but here you mention that your results are comparable to those of Ferguson et al. (2020) who observed decreases in denitrification. Please clarify that your results, with the exception of MudH, also present decreases in denitrification.

**Reply**: This has been clarified by adding this distinction to lines 334-335: *"Though the key role of redox microniches is not directly investigated here, we acquired decreases in denitrification rates in a similar range in all sediments besides MudH, especially when trawling frequencies increased."*

**20. Reviewer's comment:** Line 361. Change "resuspended silt" to "resuspended fine-grained sediment", since both silt and clay can be resuspended and preferentially advected away from bottom trawling grounds (Martin et al., 2014, PLoS ONE). You focus on how this kind of effect varies with different bottom trawling gear types, but could you elaborate on how this would affect sediment biogeochemistry?

**Reply:** We have changed the wording to "fine-grained material". The granulometry plays an important role in the regulation of sediment biogeochemistry. The grainsize distribution, permeability (the capacity to transmit fluid, dependent on the grain size), and porosity (the water content in a given volume of bulk sediment), influence the redox zonation patterns in which microbially mediated mineralization processes occur (Probandt et al., 2017), by providing binding sites for organic matter (Mayer, 1994) and by regulating the exchange of oxygen, organic matter and other reactants between the sediment and the overlying water (Huettel et al., 2014). Large, connected pore-spaces in coarse grained, permeable sediments allow for advective flows, resulting in high organic matter turnover rates fuelled by a high oxygen supply (Huettel and Rusch, 2000; Ehrenhauss et al., 2004). In contrast, cohesive sediments are characterised by smaller and less connected interstitial spaces, so that diffusive processes, molecular diffusion or transport stimulated by benthic fauna, dominate solute and particle transport (Huettel et al., 2014). A shift in grainsize to either finer sediments through settling of advected fine-grained sediments (fining), or coarsening (larger median grainsize) will thus change the biogeochemical regime in an area.

This has been implemented in the rewritten section 4.3., please see our reply to remark #1.

**21. Reviewer's comment:** Line 367-368. Modify "deep and shallow penetrating gear types", since 3 cm of penetration is not considered to be deep.

**Reply:** This sentence has now been relocated to the beginning of section 4.3, and now reads: "*In our work the differences with respect to organic matter mineralization dynamics between gear types with differing penetration depths (e.g. 3.2 ± 1.2 cm vs. 1.6 ± 0.6 cm in mud) mostly remained suggestive, rather than statistically conclusive. The fishing gear type was only included as a significant predictor for the denitrification rates, with a small coefficient (-0.00038; Table 5).*"

**22. Reviewer's comment:** Table 2: Please provide definition of the acronyms (MGS, SD.9, SD.1, VFines). Could you also give the proportion of sand (2 mm – 63 _m) content (%)? Provide also the grain size ranges to define the proportion of silt and fine-grained sediment. If finegrained sediment is consistent with the grain size range of clays (< 4 _m), please consider giving it this terminology.

**Reply:** This comment was implemented by extending the table caption with the definitions, and by updating the table itself with the % sand, and the size classes sand and mud represent. Please see the new table below, but note our reply to comment #2, for choosing to keep the references to the manuscripts in review.:

**Table 1: Characteristics of the selected sites. Low and High nutrient classification is based on relative differences in nutrient build-up for the same sediment type (see figure S1 in supplement). MGS, SD.1, and SD.9 are median grainsize, and the boundaries of the 10th and 90th percentile of the grainsize respectively (in μm). Percentage of sand (63 – 1000 μm), and mud (< 63 μm), are weight percentages of a dried sediment sample sieved over a 1 mm sieve.**

| Sediment type | Nutrients | Name | Lat (Deg. N) | Lon (Deg. E) | Depth (m) | MGS (μm) | SD.1 (μm) | SD.9 (μm) | Sand (%) | Mud (%) | Source* |
|---|---|---|---|---|---|---|---|---|---|---|---|
| Coarse sand | Low | Coarse | 51.43483 | 2.809822 | 22 | 433 ± 43 | 286 ± 31 | 660 ± 67 | 99 ± 1 | 0 ± 0 | a |
| Fine sand | Low | FineL | 55.17374 | 3.161264 | 26 | 216 ± 2 | 143 ± 0 | 328 ± 7 | 99 ± 7 | 0 ± 0 | b |
| | High | FineH | 51.1853 | 2.7013 | 9 | 220 ± 8 | 91 ± 61 | 394 ± 67 | 81 ± 5 | 9 ± 5 | a |
| Mud | Low | MudL | 58.20097 | 0.525871 | 148 | 24 ± 1 | 4 ± 0 | 67 ± 3 | 10 ± 7 | 88 ± 1 | b |
| | High | MudH | 51.2714 | 2.905033 | 11 | 19 ± 1 | 3 ± 0 | 208 ± 28 | 25 ± 4 | 74 ± 5 | a |

**\*a: Toussaint et al., *in review*; b: De Borger et al., *in review*.**

**23. Reviewer's comment:** Figure 3: A. Consider using "Day of year" in the axis title. If this carbon flux consists of only DIC, why is it applicable to both organic and inorganic fluxes? In lines 119-120 you refer to this figure as the carbon flux. Please provide more information. B. Where did the data from this figure come from? C. According to Eq. 3, biota depletion depends on both penetration depth and sediment grain size, but only penetration depth (which varies with different gear) is shown in this figure. Please include the effect of sediment grain size in this figure, since it is a very important parameter, rather than giving it in Table 3.

**Reply:** We have amended this figure by (1) changing the main title and x-axis title of panel A, (2) changing the positions of panel B and C, in accordance with the text. The penetration depth per sediment type originates from Pitcher (pers. comm), please see our reply to remark #12. In terms of showing all decreases of bioturbation, we feel that this example panel (Figure 2 B), in combination with Table 4 (note the numbering has changed) and Eq. 3 should suffice.

**24. Reviewer's comment:** Consider converting Tables 5 and 7 as figures, since the changes in terms of frequency, gear type, and sediment type would be more visual.

**Reply:** Tables 5 and 7 have been moved to the supplementary material (now resp. Table S1 and S2). In place of Table 5 is now a slightly adapted version of Figure S2, previously in the supplement, showing averaged nutrient profiles per trawling intensity for each location. There is now also a figure to replace Table 7 (please see below), that shows the same information graphically. To make space for these figures, we chose to remove Figure 4, the range in porewater concentrations throughout the year, to the supplement.

[Figure]

**Figure 1: Rates of total mineralization (A - E), and the three main mineralization processes (F - J: oxic; K - O: anoxic; P - T: denitrification) (y-axis, mmol m$^{-2}$ d$^{-1}$) for each gear type (blue boxes: tickler gear; red boxes: pulse gear), and for increasing trawling frequency (x-axis, y$^{-1}$)**

**25. Reviewer's comment:** Figure S1: The fitted red lines for FineL-oxygen, MudH-nitrate are not adjusted to the concentration profiles measured.

**Reply:** For FineL there was no oxygen profile to fit the model to, as we did not want to risk damaging the oxygen sensor on visibly present shell fragments in this sediment on the second day into a 15 day research cruise. The MudH NO$_3^-$ profiles could not be fitted without simultaneously compromising deep NH$_4^+$ concentrations, with the latter more representative of nutrient build-up with increasing depth. Additionally, we believe it is likely that the NO$_3^-$ concentrations represent a part of the considerable NH$_4^+$ content that has been oxidized during experimental procedures.

**REVIEWER #2**

395    **1. Reviewer's comment:** Despite this, I guess that the authors should make an effort to better explain how the bioturbation rates were calculated. As far as I can see, this information is not fully included in the manuscript and this could be crucial to understand how much and whether the abundance, biomass and (functional) diversity of the benthos in the parameterization sites have been taken into account. In my opinion, this
400    information is also necessary because the results, though contextualized according to the different sediment granulometry of the experimental sites and C fluxes, do not seem to have been analyzed eliminating the covariate effect of water depth, which controls benthos abundance, biomass and diversity, which, in turn, could respond differently to bottom trawling disturbance (as indeed postulated in the discussion).

405    **Reply:** A more specific description of how the biodiffusivity constant $Db$ (~bioturbation rate) was estimated based on porewater nutrient profiles has been added in the methodology section, as follows: "*bottom water, as well as process rate parameters that were derived following a 2-step steady state fitting procedure (Table 3). Using the measured DIC flux as the upper boundary organic carbon input flux, the $O_2$ flux and porewater profiles of $O_2$, $NO_3^-$, and $NH_4^+$ were first fitted manually by tweaking a limited set of model parameters. The degradation rate of refractory material (rSlow), and the biodiffusivity constant Db were constrained by fitting*
410    *$NH_4^+$, and $O_2$ profiles. Mechanistically, decreasing the bioturbation rate Db reduces the build-up of $NH_4^+$ with depth, increases the oxygen penetration depth, and changes the shape of the $NO_3^-$ profile (deepening the $NO_3^-$ peak).*" This is thus a pure biogeochemical approach to bioturbation, in which the biodiffusivity constant is assumed to be the product of indeed biomass, abundance, and functional diversity, but the latter are not considered separately. However, this is indeed important information to consider when deciding on real-case
415    management steps, as a variety of studies have indicated the role of life-history and behavioral traits of individual species in their response to trawling (e.g., Tillin et al., 2006; Rijnsdorp et al., 2016; Pitcher et al., 2017; Hiddink et al., 2019).
       We performed tests to see if (part) of this information could be implemented in more detail, based on known species communities in the specific areas. The complexity of shifting species communities, unknowns about
420    individual species behaviors, and most importantly the difficulty to translate all this information to a model of sediment diagenesis, prohibited us from doing so successfully. This last step would be of high value to the field of benthic ecology and biogeochemistry in general, and requires extensive research.
       In the end, the meta-study of Sciberras et al., 2018 provided a "blanket" formula that could be used, assuming that a culling of organism equates to a culling of bioturbation.

425

       **2. Reviewer's comment:** As mineralization rates, as correctly postulated in the manuscript, are dependent also on the relative importance of refractory and labile/semi-labile fractions of OC, it could be interesting to see addressed the effects of bottom trawling on the two fractions, though this could be the object of a "sister" manuscript.

430

       **Reply:** This is (shortly) mentioned on lines 208 – 213, "*A redistribution of organic carbon was visible in the upper cm of the sediment, where organic carbon concentrations were higher in the impacted than in the baseline simulation (example in the cutout of the top 5 mm shown for MudH, Figure 5). In FineL, MudL and MudH the ratio of labile organic carbon (FDET) to semi-labile organic carbon (SDET) increased between*
435    *25 and 34 % (Figure S3, supplement). This effect was only noticeable in the upper 0.2 – 0.5 cm, below this depth values of this ratio in all trawling frequencies converged to 0, due to the depletion of labile organic carbon.*"
       In our model, bottom trawling causes a depletion of the carbon build-up in the sediment. As organic carbon generally becomes less reactive with age (~ depth in the sediment), the total organic carbon pool becomes
440    more reactive when less organic carbon gradually builds up to deeper layers. In the most extreme case (as an example), there is almost no organic carbon in the sediment due to continuous, chronic resuspension. If fresh organic carbon is then deposited on the sediment-water interface, the labile organic carbon will proportionally make up a considerable part of the total organic carbon pool.
       Since also the other reviewer noted interest in changes to the reactive carbon pool, we added the figure (see
445    below) to the supplementary information, and refer to this figure on line 235: "*In FineL, MudL and MudH the ratio of labile organic carbon (FDET) to semi-labile organic carbon (SDET) increased between 25 and 34 % (Figure S3, supplement).*"

       We choose not to include the specific effect of this shift towards more labile organic carbon near the surface
450    as a topic for extended discussion, for 2 main reasons: (1) the drastic changes to the total organic carbon pool and availability of reactants in general have a far greater effect on mineralization processes. (2) The significance to this pertains more to the temporal (sub-annual) dynamics of organic matter mineralization,

which was consciously left out of this manuscript. It would indeed be more suited for a sister-manuscript, in which seasonal dynamics and management implications thereof are discussed.

455

[Figure]

**Figure S 2: Annually averaged modelled quality of the reactive organic carbon pool in the surface sediments (note different depths on y-axis between figures for visualization purposes). The carbon quality (x-axis) is represented as the proportion of fast degrading detritus (FDET, labile org. C) in the summed labile and semi-labile org. C pool (FDET + SDET). Black dotted line is the 0 trawl default, full and dotted coloured lines are tickler and pulse gear respectively, with increasing trawling frequencies as different colours.**

3. **Reviewer's comment:** Technical suggestions: Despite all of the figures are necessary and informative, a (qualitative) graphic panel of differences and tendencies of mineralization rates along the trawling frequency gradient in the different environmental contexts (sediment type and depth) would help a lot the general reader to recap the results.

465

**Reply:** We have implemented this change by replacing tables 5 and 7 with graphical alternatives, and suggesting a summary-type figure as the graphical abstract.

Given the nuance in some of the results (e.g. changes to denitrification, initial decreases and then increases in porewater solutes and vice-versa, this would again become a quite large figure. To compromise, we suggest to use the following figure as the figure for the abstract. This figure shows the changes to total, and relative mineralization rates at 1 and 5 trawls $y^{-1}$. Since there is little/no difference between gear types, this is based on the tickler chain gear results.

[Figure]

475

4.  **Reviewer's comment:** The number of tables could be reduced, moving some of them in the supplement material. Figures' and Tables' numbering (and their order of reference in the main text) need an accurate check and correction.

480    **Reply:** This has been implemented as follows. Tables 5 and 7 have been moved to the supplementary material (now resp. Table S1 and S2). In place of Table 5 is now a slightly adapted version of Figure S2, previously in the supplement, showing averaged nutrient profiles per trawling intensity for each location. There is now also a figure to replace Table 7 (please see below), that shows the same information graphically. To make space for these figures, we chose to remove Figure 4, the range in porewater concentrations throughout the year, to 485    the supplement.

[Figure]

**Figure 2: Rates of total mineralization (A - E), and the three main mineralization processes (F - J: oxic; K - O: anoxic; P - T: denitrification) (y-axis, mmol m$^{-2}$ d$^{-1}$) for each gear type (blue boxes: tickler gear; red boxes: pulse gear), and for increasing trawling frequency (x-axis, y$^{-1}$). Note different scales on y-axes.**

490

**Added text to supplement:**

**S.1. Data collection for parametrization**

495

**S.1.1. General sampling design**

In September 2017, locations Coarse, FineH, and MudH were sampled in the Belgian Part of the North Sea (Toussaint et al., *in review*), whereas FineL and MudL were sampled in the Central-Northern North Sea in May-June of 2018 (De Borger et al., *in review*). A stainless steel NIOZ boxcorer was used to sample the sediments used

500   to describe the different locations in this modelling study (30 cm ID, 50 cm height). At each location, triplicate intact boxcores were collected.

From the September 2017 samples, a set of subcores was taken from each boxcore sample for: incubation purposes (Ø 19 Plexiglass sampling cores for coarse grained sediment to allow for a stirring mechanism for advective flows; Ø 10 cm for cohesive sediment, 10-15 cm deep + 10 cm of overlying water), to determine porewater nutrient

505   profiles (Ø 10 cm Plexiglass sampling core), and to determine sediment characteristics (cut off syringe, upper 3 cm). Incubations were performed in the dark (to prevent photosynthetic activity), and in climate controlled laboratory conditions with disk (coarse) of teflon (cohesive) stirrers agitating the overlying water, and exchange rates of oxygen, dissolved inorganic carbon (DIC), and dissolved inorganic nitrogen (DIN) were measured.

In the May-June samples, ship-board incubations (dark) were performed using the entire boxcore sample,

510   measuring the same parameters as previously mentioned. For this, the boxcore "bucket" containing the sediment was sealed with a Plexiglass lid containing a Teflon stirrer, and placed in a buffering vat on deck to maintain steady temperature. After this shipboard incubation, subcores were collected to measure porewater nutrient profiles (Ø 10 cm Plexiglass sampling core), oxygen microprofiles (Ø 5 cm Plexiglass sampling core), and sediment characteristics (cut off syringe, upper 2 cm).

515

**S.1.2. Flux calculations**

During incubations, the oxygen concentration in the overlying water was monitored using optode sensors (FirestingO2, Pyroscience, 2-point calibration), set at 1 Hz. At the same time, DIC and DIN concentrations were sampled from the overlying water with syringes at discreet time intervals. 5 – 10 mL were collected for DIN, and

520   filtered through a 0.45 µm syringe filter, and stored at -20 °C until further processing. 6 - 10 mL of water were collected in headspace vials for DIC, and subsequently poisoned with 1 µL of saturated HgCl2 per mL sample for preservation and kept refrigerated at 4 °C until further processing. During incubations, $O_2$ concentrations did not decrease below 50 % of the initial oxygen concentration. As such, incubations in the 2017 samples lasted between 2 – 8 hours, and 24 – 36 hours in 2018.

525   Upon thawing, nutrient concentrations were determined by a SEAL QuAAtro segmented flow analyser (Jodo et al., 1992). DIC analysis was performed using a segmented flow analyser (San++ SKALAR) following (Stoll et al., 2001). Fluxes (in mmol $m^{-2}$ $d^{-1}$) were calculated by fitting a linear regression through the concentration time series, and multiplying the regression coefficient by the height of the overlying water to convert from volumetric to surficial rates. For oxygen fluxes the same method was applied to a consistently decreasing section of the oxygen

530   concentration data.

**S.1.3. Porewater nutrient profiles**

Porewater nutrients (DIN) were collected in 1-2 cm interval depth slices down to 12 cm deep, using rhizon samplers (0.15 µm pore size, Rhizosphere Research Products). The rhizons were inserted into the sediment core

535   through pre-drilled holes in the core wall, and a maximum of 4 mL of porewater was extracted from each interval using a 5 mL syringe connected to the rhizon sampler (Seeberg-Elverfeldt et al., 2005; Dickens et al., 2007; Shotbolt, 2010). Further processing of the nutrient samples was done the same as for the nutrient flux samples.

**S.1.4. Oxygen microprofiles**

540   Oxygen-depth profiles in the sediment were measured using Clark-type $O_2$ micro-electrodes (50 µm tip diameter, Unisense) (Revsbech, 1989). Readings were taken at 100 µm intervals, starting 2000 µm (2 mm) above the sediment-water interface (water aerated to 100% $O_2$ saturation before the experiment) down to the depth in the sediment at which all oxygen was depleted. A two-point calibration was conducted prior to measurements using 100 and 0 % oxygen saturated seawater to represent water column and anoxic $O_2$ concentrations, respectively. In

545   each sediment core, up to three replicate profiles were taken from different areas of the sediment (except in Coarse, where the risk of damage to the sensor due to coarseness of the sediment was determined too great).

**S.1.5. Sediment characteristics**

Sediment grain size was determined by laser diffraction on freeze-dried and sieved (< 1 mm) sediment samples in

550   a Malvern Mastersizer 2000 (McCave et al., 1986). Grain size fractions were determined as volume percentages according to the Wentworth scale (Wentworth, 1922): clay/silt (< 63 µm), very fine sand (vfines: 63 – 125 µm),

fine sand (fines: 125 – 250 μm), medium sand (250 – 500 μm), and coarse sand (500 μm – 1 mm). In this manuscript, the percentage of sand was calculated by summing grainsize classes between 63 and 1000 μm. The median grain size (MGS) was calculated on the fraction < 1 mm. Water content was determined as the volume of water removed by freeze drying wet sediment samples. The sediment density was determined by measuring the water displacement of a given weight of dried sediment. Sediment porosity was determined from water content and solid phase density measurements, accounting for the salt content of the pore water.

**Additional references**

de Jonge, D. S. W., Stratmann, T., Lins, L., Vanreusel, A., Purser, A., Marcon, Y., et al. (2020). Abyssal food-web model indicates faunal carbon flow recovery and impaired microbial loop 26 years after a sediment disturbance experiment. *Prog. Oceanogr.* 189, 102446. doi:10.1016/j.pocean.2020.102446.

Depestele, J., Degrendele, K., Esmaeili, M., Ivanovic, A., Kröger, S., O'Neill, F. G., et al. (2019). Comparison of mechanical disturbance in soft sediments due to tickler-chain SumWing trawl vs. Electro-fitted PulseWing trawl. *ICES J. Mar. Sci.* 76, 312–329. doi:10.1093/icesjms/fsy124.

Dickens, G. R., Koelling, M., Smith, D. C., and Schnieders, L. (2007). Rhizon sampling of pore waters on scientific drilling expeditions: An example from the IODP expedition 302, Arctic Coring Expedition (ACEX). *Sci. Drill.*, 22–25. doi:10.2204/iodp.sd.4.08.2007.

Dounas, C. G., Davies, I. M., Hayes, P. J., Arvanitidis, C. D., and Koulouri, P. T. (2005). The effect of different types of otter trawl ground rope on benthic nutrient releases and sediment biogeochemistry. *Benthic Habitats Eff. Fish.* 41, 539–544.

Ehrenhauss, S., Witte, U., Janssen, F., and Huettel, M. (2004). Decomposition of diatoms and nutrient dynamics in permeable North Sea sediments. *Cont. Shelf Res.* 24, 721–737. doi:10.1016/j.csr.2004.01.002.

Ferguson, A. J. P., Oakes, J., and Eyre, B. D. (2020). Bottom trawling reduces benthic denitrification and has the potential to influence the global nitrogen cycle. *Limnol. Oceanogr. Lett.* doi:10.1002/lol2.10150.

Gausepohl, F., Hennke, A., Schoening, T., Köser, K., and Greinert, J. (2020). Scars in the abyss: reconstructing sequence, location and temporal change of the 78 plough tracks of the 1989 DISCOL deep-sea disturbance experiment in the Peru Basin. *Biogeosciences* 17, 1463–1493. doi:10.5194/bg-17-1463-2020.

Huettel, M., Berg, P., and Kostka, J. E. (2014). Benthic Exchange and Biogeochemical Cycling in Permeable Sediments. 23–51. doi:10.1146/annurev-marine-051413-012706.

Huettel, M., and Gust, G. (1992). Impact of bioroughness on interfacial solute exchange in permeable sediments. *Mar. Ecol. Prog. Ser.* 89, 253–267. doi:10.3354/meps089253.

Huettel, M., and Rusch, A. (2000). Transport and degradation of phytoplankton in permeable sediment. *Limnol. Oceanogr.* 45, 534–549. doi:10.4319/lo.2000.45.3.0534.

ICES (2020). Working Group on Electrical Trawling (WGELECTRA). *ICES Sci. Reports / Rapp. Sci. du Ciem* 1, 87. doi:10.17895/ices.pub.5619.

Jodo, M., Kawamoto, K., Tochimoto, M., and Coverly, S. C. (1992). Determination of nutrients in seawater by segmented-flow analysis with higher analysis rate and reduced interference on ammonia. *J. Automat. Chem.* 14, 163–167. doi:10.1155/S1463924692000300.

Martín, J., Puig, P., Masqué, P., Palanques, A., and Sánchez-Gómez, A. (2014). Impact of bottom trawling on deep-sea sediment properties along the flanks of a submarine canyon. *PLoS One* 9. doi:10.1371/journal.pone.0104536.

Mayer, L. M. (1994). Surface area control of organic carbon accumulation in continental shelf sediments. *Geochim. Cosmochim. Acta* 58, 1271–1284. doi:10.1016/0016-7037(94)90381-6.

McCave, I. N., Bryant, R. J., Cook, H. F., and Coughanowr, C. A. (1986). Evaluation of a laser-diffraction-size analyzer for use with natural sediments. *J. Sediment. Res.* 56, 561–564. doi:10.1306/212f89cc-2b24-11d7-8648000102c1865d.

McConnaughey, R. A., Hiddink, J. G., Jennings, S., Pitcher, C. R., Kaiser, M. J., Suuronen, P., et al. (2020). Choosing best practices for managing impacts of trawl fishing on seabed habitats and biota. *Fish Fish.* 21, 319–337. doi:10.1111/faf.12431.

Mengual, B., Cayocca, F., Le Hir, P., Draye, R., Laffargue, P., Vincent, B., et al. (2016). Influence of bottom trawling on sediment resuspension in the 'Grande-Vasière' area (Bay of Biscay, France). *Ocean Dyn.* 66, 1181–1207. doi:10.1007/s10236-016-0974-7.

Mengual, B., Le Hir, P., Cayocca, F., and Garlan, T. (2019). Bottom trawling contribution to the spatio-temporal variability of sediment fluxes on the continental shelf of the Bay of Biscay (France). *Mar. Geol.* 414, 77–91. doi:10.1016/j.margeo.2019.05.009.

Morato, T., Watson, R., Pitcher, T. J., and Pauly, D. (2006). Fishing down the deep. *Fish Fish.* 7, 24–34. doi:10.1111/j.1467-2979.2006.00205.x.

Murray, F., Copland, P., Boulcott, P., Robertson, M., and Bailey, N. (2016). Impacts of electrofishing for razor clams (Ensis spp.) on benthic fauna. *Fish. Res.* 174, 40–46. doi:10.1016/j.fishres.2015.08.028.

Norse, E. A., Brooke, S., Cheung, W. W. L., Clark, M. R., Ekeland, I., Froese, R., et al. (2012). Sustainability of deep-sea fisheries. *Mar. Policy* 36, 307–320. doi:10.1016/j.marpol.2011.06.008.

Paradis, S., Pusceddu, A., Masqué, P., Puig, P., Moccia, D., Russo, T., et al. (2019). Organic matter contents and degradation in a highly trawled area during fresh particle inputs (Gulf of Castellammare, southwestern Mediterranean). *Biogeosciences* 16, 4307–4320. doi:10.5194/bg-16-4307-2019.

Poos, J.-J., Hintzen, N. T., van Rijssel, J. C., and Rijnsdorp, A. D. (2020). Efficiency changes in bottom trawling for flatfish species as a result of the replacement of mechanical stimulation by electric stimulation. *ICES J. Mar. Sci.* doi:10.1093/icesjms/fsaa126.

Price, W. L. (1977). A controlled random search procedure for global optimisation. *Comput. J.* 20, 367–370. doi:10.1093/comjnl/20.4.367.

Probandt, D., Knittel, K., Tegetmeyer, H. E., Ahmerkamp, S., Holtappels, M., and Amann, R. (2017). Permeability shapes bacterial communities in sublittoral surface sediments. *Environ. Microbiol.* 19, 1584–1599. doi:10.1111/1462-2920.13676.

Puig, P., Canals, M., Company, J. B., Martín, J., Amblas, D., Lastras, G., et al. (2012). Ploughing the deep sea floor. *Nature* 489, 286–289. doi:10.1038/nature11410.

Pusceddu, A., Bianchelli, S., Martín, J., Puig, P., Palanques, A., Masqué, P., et al. (2014). Chronic and intensive bottom trawling impairs deep-sea biodiversity and ecosystem functioning. *Proc. Natl. Acad. Sci. U. S. A.* 111, 8861–8866. doi:10.1073/pnas.1405454111.

Revsbech, N. P. (1989). An oxygen microsensor with a guard cathode. *Limnol. Oceanogr.* 34, 474–478. doi:10.4319/lo.1989.34.2.0474.

Sciberras, M., Hiddink, J. G., Jennings, S., Szostek, C. L., Hughes, K. M., Kneafsey, B., et al. (2018). Response of benthic fauna to experimental bottom fishing: A global meta-analysis. *Fish Fish.* 19, 698–715. doi:10.1111/faf.12283.

Seeberg-Elverfeldt, J., Schluter, M., Feseker, T., and Kolling, M. (2005). Rhizon sampling of porewaters near the sediment-water interface of aquatic systems. *Limnol. Oceanogr.* 3, 361–371. doi:Pii S0012-821x(02)01064-6 Doi 10.1016/S0012-821x(02)01064-6.

Shotbolt, L. (2010). Pore water sampling from lake and estuary sediments using Rhizon samplers. *J. Paleolimnol.* 44, 695–700. doi:10.1007/s10933-008-9301-8.

Soetaert, K., and Petzoldt, T. (2010). Inverse Modelling, Sensitivity and Monte Carlo analysis in R Using PAckage FME. *J. Stat. Softw.* 33, 1–28. doi:10.18637/jss.v033.i03.

Soetaert, M., Chiers, K., Duchateau, L., Polet, H., Verschueren, B., and Decostere, A. (2015). Determining the safety range of electrical pulses for two benthic invertebrates: brown shrimp (Crangon crangon L.) and ragworm (Alitta virens S.). *ICES J. Mar. Sci.* 72, 973–980. doi:10.1093/icesjms/fsu176.

Soetaert, M., Verschueren, B., Chiers, K., Duchateau, L., Polet, H., and Decostere, A. (2016). Laboratory Study of the Impact of Repetitive Electrical and Mechanical Stimulation on Brown Shrimp Crangon crangon. *Mar. Coast. Fish.* 8, 404–411. doi:10.1080/19425120.2016.1180333.

Stoll, M. H. C., Bakker, K., Nobbe, G. H., and Haese, R. R. (2001). Continuous-flow analysis of dissolved inorganic carbon content in seawater. *Anal. Chem.* 73, 4111–4116. doi:10.1021/ac010303r.

Tiano, J. C., Witbaard, R., Bergman, M. J. N., Van Rijswijk, P., Tramper, A., Van Oevelen, D., et al. (2019). Acute impacts of bottom trawl gears on benthic metabolism and nutrient cycling. *ICES J. Mar. Sci.* 76, 1917–1930. doi:10.1093/icesjms/fsz060.

Trimmer, M., Petersen, J., Sivyer, D., Mills, C., Young, E., and Parker, E. (2005). Impact of long-term benthic trawl disturbance on sediment sorting and biogeochemistry in the southern North Sea. *Mar. Ecol. Prog. Ser.* 298, 79–94. doi:10.3354/meps298079.

van Marlen, B., de Haan, D., van Gool, A., and Burggraaf, D. (2009). The effect of pulse stimulation on marine biota – Research in relation to ICES advice – Progress report on the effects on benthic invertebrates. 53.

Vonnahme, T. R., Molari, M., Janssen, F., Wenzhöfer, F., Haeckel, M., Titschack, J., et al. (2020). Effects of a deep-sea mining experiment on seafloor microbial communities and functions after 26 years. *Sci. Adv.* 6, eaaz5922. doi:10.1126/sciadv.aaz5922.

Watson, R. A., and Morato, T. (2013). Fishing down the deep: Accounting for within-species changes in depth of fishing. *Fish. Res.* 140, 63–65. doi:10.1016/j.fishres.2012.12.004.

Wentworth, C. K. (1922). A Scale of Grade and Class Terms for Clastic Sediments. *J. Geol.* 30, 377–392. doi:www.jstor.org/stable/30063207.

---

## Author Response (AR2)

We thank the reviewer for the useful suggestions for further improvements, which we have implemented for the most part. We also suggest areas of further research, such as implementing this kind of model in deep-sea/margin settings. We also updated the references to what was initially unpublished work throughout the manuscript: Toussaint et al., 2021, and De Borger et al., 2021 have been published during this reviewing process.

Changes have been implemented in the manuscript as track changes, and copied here in our reply. With these changes, we hope to have addressed the remarks sufficiently.

**Main comments:**

1. **Reviewer comment:** Abstract: Since penetration depth of tickler chain and electric pulse trawl are not significantly different (Fig. 2C), please refrain from using "deeper" and "shallower" descriptions of both gear types in the abstract. Instead, I would suggest simply stating "two gear types with contrasting degrees of disturbance of the seafloor".

   **Reply:** We prefer not to alter this wording anymore (it has been previously changed from deep/shallow to deeper/shallower). The different degree of disturbance is specifically due to the differences in depth and variability, so it would be strange not to mention it immediately. Because of the used variance around the mean penetration of both gears, differences may not be statistically significant as implied by the figure, but still depths have been set 50 % lower for the shallower gear type.

2. **Reviewer comment:** Line 118-119: Using the measured DIC flux as the upper boundary organic carbon input flux. How can the measured DIC flux be used to estimate OC input? When you refer to OC flux, do you refer to particulate OC or dissolved OC? This sinusoidal carbon flux was estimated using what data? Is it a real representation of carbon input to the seafloor? Why use an amplitude of 1?

   **Reply:** Assuming (quasi-) steady state with little to no burial of OC (valid for a lot of areas in the North Sea), we expect the carbon output (DIC) to be similar to the input of organic matter, as in the sediment this organic matter is mineralised to DIC which fluxes back to the water column. The OC flux refers to the particulate OC flux, as in organic detritus arriving on the seabed. Choosing a sinusoidal function is a model simplification which represents the input of carbon to the sediment in shallow North Sea sediments reasonably well, as seen in this example below from Provoost et al., 2013, *Estuarine, Coastal and Shelf Science*. The amplitude of 1 causes a fluctuation around the steady state-mean carbon input, which touches 0 in winter. We have added clarification and explanation of our reasoning to lines 138-141: "*A sinusoidally varying detrital carbon deposition flux, with the model derived carbon flux (Cflux, Table 3) as the annual average, and imposing an amplitude of 1 was used as the upper boundary organic carbon flux (Figure 2 A). The uniform amplitude of 1 for all sites was chosen to simplify temporal variations between sites.*"

[Figure]

Fig. 3. Measurements of water column chlorophyll-a (a), temperature (b), sediment chlorophyll-a (c) and benthic oxygen consumption (d). Values of water column chlorophyll-a and temperature were interpolated to be used as forcing functions for the model. For (c) and (d) the global model sensitivity is given as well, with light grey indicating the range of model results and dark grey the standard deviation. The dashed line shows the median as well as the best-fit model run, which cannot be distinguished visually.

3.  **Reviewer comment:** The varying OC flux in the different stations is not addressed in the methods. In lines 137 – 139. "A sinusoidally varying carbon deposition flux with the model derived carbon flux (Cflux, Table 3) as the annual average, and imposing an amplitude of 1 was used as the upper boundary organic carbon flux (Figure 2 A). This resulted in differing organic carbon fluxes for each location." However, this refers to the temporal variations in OC flux, and not the spatial variations. Could you please specify how you obtained the OC flux stated in Table 3?

    **Reply:** This has been explained in the methodology (lines 117-120): "*Model parameters included both measured concentrations in the bottom water, as well as process rate parameters that were derived following a 2-step steady state fitting procedure (Table 3). Using the measured DIC flux as the upper boundary organic carbon input flux, the $O_2$ flux and porewater profiles of $O_2$, $NO_3^-$, and $NH_4^+$ were first fitted manually by tweaking a limited set of model parameters.*" The description of how the DIC flux was measured, is in the supplementary information under section S1.2. Please see our previous reply on how we further addressed the varying OC flux.

4.  **Reviewer comment:** At the beginning of the discussion (4.1 Organic carbon depletion) you mention that trawling removes OC through erosion, and that less OC is redistributed due to the absence of bioturbation. However, bottom trawling also mixes up sediment, which you consider in section 2.1.3 Disturbance modelling. Please include all of these aspects in this section of the discussion: trawling-derived erosion, mixing, and reduced bioturbation. In your disturbance simulation, the effect of the removal of bioturbators is far greater than the sediment mixing effect of bottom trawlers, which I consider is an important aspect to include in this section. To further add to this discussion, would the sediment mixing effect caused by bioturbators and bottom trawling be similar with an increasing trawling frequency? In certain fishing grounds, bottom trawling often occurs on a daily basis (i.e. Catalan margin).

    **Reply:**

    We have rewritten the first paragraph to better include the mixing event itself, though the effects of this (injection of $O_2$, $NO_3^-$ to deeper layers) remain mostly discussed in 4.2. Lines 287-296 now read: "*The main drivers of the biogeochemical changes were found to be the depletion of organic carbon (OC) in the sediment (i.e. the substrate for mineralization itself), the redistribution of this OC nearer to the SWI (Figure 5), and the increasing oxygenation of the sediment. With each trawl pass, a part of the organic carbon rich top layer is removed (Figure 5). This is associated with an*

*injection of oxidized reactants from the water column (O₂, NO₃ ) deeper into the sediment, and a homogenization of OC concentrations in the mixed layer during the mixing phase. Simultaneously, part of the benthos in the sediment is removed, often strongly decreasing the bioturbation rate, affecting the rate at which organic matter is distributed in the sediment (especially after multiple trawling events y⁻¹, Table 4). Sediment mixing alone could potentially increase OC contents at the bottom of the mixing zone, but successive trawling events, and the removal of bioturbators that can transport OC far below the mixing zone, resulted in a redistribution of OC closer to the sediment-water interface in all simulated sediments.*"

We further point out that it is too general to say that the removal of bioturbators had a far greater effect that the mixing by the trawling gear. The mixing by the trawling gear clearly affects mineralization processes as well, among others by spiking sediment $O_2$ or $NO_3^-$ concentrations which affected mineralization process rates up to weeks. However, our bioturbating organisms did have a potential greater effect on the distribution of OC, as has now been put in perspective in the previous paragraph: "*Sediment mixing alone could potentially increase OC contents at the bottom of the mixing zone, but successive trawling events, and the removal of bioturbators that can transport OC far below the mixing zone, resulted in a redistribution of OC closer to the sediment-water interface in all simulated sediments.*"

Whereas your last suggestion is interesting, this is not something we have explored in this iteration of the model development, and adding speculation to the discussion might not be meaningful. My personal intuition is that mixing on a daily basis will make the trawling gear by far the dominant mixing process (without any knowledge of the benthos on the Catalan margin).

5. **Reviewer comment:** The effect of grain size and permeability in mineralization pathways are mainly described in section 4.2 Changes to mineralization pathways, lines 345-358. However, these effects are, in my opinion, unnecessarily repeated in section 4.3 Reducing gear penetration depth, lines 389-400. I highly suggest to discuss the effect of grain size and permeability only in section 4.2, and avoid going into detail in section 4.3, since this complicates the comprehension of the latter section. Moreover, I would suggest to change the section title to "Reducing gear disturbance on the seafloor", since it more appropriately describes the content of the section.

    **Reply: T**here was indeed unnecessary repetition of this explanation, we have removed lines 398-403 from section 4.3. The section now flows from the importance of disturbing only a thin surface layer, to changes to sediment structure (Lines 390-394): "*… This indicates that only a thin layer of surface sediment needs to be disturbed to generate significant biogeochemical changes (Dounas et al., 2005). Many biogeochemical processes are mediated by the dynamics of oxygen near the sediment-water interface, which itself is influenced by the composition, and permeability of the sediment. A shift towards fining (an increased proportion of finer grain size classes) has been described in certain trawled areas, with expected consequences for sediment biogeochemistry, such as an increased rate of sulphate reduction (Trimmer et al., 2005). ...*"

    We moved the reference to Huettel and Gust 1992 to section 4.2., where lines 355-357 now read: "*Larger pore-spaces in these sediments allow for bottom water to penetrate more deeply into the sediment matrix, bringing oxygen and other reactants into deeper sediment layers (Huettel and Gust, 1992). Cohesive sediments mostly lack such advective transport.*"

    We have also changed the title of section 4.3 to "*Reducing gear disturbance of the seafloor*" as per your suggestion.

6. **Reviewer comment:** Lines 421-422. A better citation to "Site-specific conditions such as rates of biogeochemical recovery and sedimentation rates need to be known to determine the resilience of ecosystems to trawling, and fine-tune management plans" would be Paradis et al. (2021),

Geophysical Research Letters, that addresses the effect of seasonal trawling closures in organic matter content of trawling grounds combined with the sedimentological characteristics.

**Reply:** Thank you for this interesting reference, we have replaced it.

7. **Reviewer comment:** Code availability. Please upload your code and data to Github or a similar repository.

**Reply:** We will put the R-package that contains the diagenetic model on R-forge and will publish the code that performs the simulation in a Github repository upon publication of the manuscript.

**Technical corrections**

8. **Reviewer comment:** Line 53-54: "Observed biogeochemical changes caused by sediment resuspension can lead to the instantaneous release of nutrients from the sediment into the water column". Please add a reference to this release of nutrients from the sediment into the water column. For instance, one of the findings of the INTERPOL project published by Durrieu de Madron et al., 2005, Continental Shelf Research, studies this.

**Reply:** Thank you for this suggestion, we have added a reference to Durrieu de Madron et al. (2005) to the sentence.

9. **Reviewer comment:** In line 225, "In FineL, MudL and MudH the ratio of labile organic carbon (FDET) to semi-labile organic carbon (SDET) increased between 25 and 34 % (Figure S3, supplement)". However, Figure S3 actually refers to FDET / (FDET + SDET). Moreover, in lines 83-84, you define SDET as slow decaying fraction, while here it is defined as semi-labile organic carbon (as well as in line 236). Please maintain consistency in the nomenclature.

**Reply:** We have replaced the word "refractory" with "semi-labile" on lines 84 and 124 to remain consistent with that this component represents. Secondly, we updated line 237 to: "*In FineL, MudL and MudH the ratio of labile organic carbon (FDET) as a proportion of the carbon pool (FDET + SDET) increased between 25 and 34 % (Figure S3, supplement).*"

10. **Reviewer comment:** Lines 207-210. Please add the mineralization pathways of the Coarse sediment station, as you do for the remaining stations.

**Reply:** The mineralization pathways for the Coarse sediment are given on line 204-205: "*In the coarse sand station (Coarse), the average total mineralization rate was 13.6 mmol C $m^{-2}$ $d^{-1}$, with 89 % of this due to oxic mineralization, 6 % due to anoxic mineralization, and 5 % was denitrified (see Table S2, supplement).*"

11. **Reviewer comment:** Line 262. "[…] at 4 out of 5 stations". Instead of phrasing it this way, say "at all stations except MudH", or something along those lines. It is easier for the reader to follow.

**Reply:** We have implemented this suggestion in line 262-264, which now reads: "*Denitrification rates (base: 0.6, 0.5, 0.1, 0.8, 4.2 mmol m-2 d-1 for Coarse, FineL, FineH; MudL, and MudH respectively) decreased with increasing trawling frequencies at all stations except MudH, with a maximum reduction of 74 % (tickler) and 68 % (pulse) at FineL (Figure 6: P-T).*"

12. **Reviewer comment:** Line 306. Add that these impacts could be far greater in deep-seafloor environments. Similar studies as those explained in this study should be conducted.

**Reply:** We have implemented this suggestion as follows (lines 317-319): "*All three of these factors increase the impacts of trawling events on organic matter cycling in the model presented in this*

*work, and further modelling work could be useful to investigate the potentially large impact that deep-sea habitats experience.*"

13. **Reviewer comment:** Line 313. Mengual et al., 2016 addresses trawling in shallow environments, and not in the deep-sea.

    **Reply:** This reference was used in connection to the "fine" aspect, and not as a study of resuspension in deep sediments specifically. We have reworded this sentence to: "*Deep-water species communities are slow-growing and thus recover slowly, organic matter deposition rates are low, and the generally finer-grained sediments found in the deep are easily resuspended following a trawl passage (Norse et al., 2012; Mengual et al., 2016).*"

14. **Reviewer comment:** Line 332. "(, Figure 7)". Erase the ", " when citing the figure.

    **Reply:** Thank you for noticing this, the comma has been removed.